# The pupal moulting fluid has evolved social functions in ants

Orli Snir[1✉], Hanan Alwaseem[2], Søren Heissel[2], Anurag Sharma[3], Stephany Valdés-Rodríguez[1,4], Thomas S. Carroll[5], Caroline S. Jiang[6], Jacopo Razzauti[1] & Daniel J. C. Kronauer[1,4✉]

Insect societies are tightly integrated, complex biological systems in which group-level properties arise from the interactions between individuals[1–4]. However, these interactions have not been studied systematically and therefore remain incompletely known. Here, using a reverse engineering approach, we reveal that unlike solitary insects, ant pupae extrude a secretion derived from the moulting fluid that is rich in nutrients, hormones and neuroactive substances. This secretion elicits parental care behaviour and is rapidly removed and consumed by the adults. This behaviour is crucial for pupal survival; if the secretion is not removed, pupae develop fungal infections and die. Analogous to mammalian milk, the secretion is also an important source of early larval nutrition, and young larvae exhibit stunted growth and decreased survival without access to the fluid. We show that this derived social function of the moulting fluid generalizes across the ants. This secretion thus forms the basis of a central and hitherto overlooked interaction network in ant societies, and constitutes a rare example of how a conserved developmental process can be co-opted to provide the mechanistic basis of social interactions. These results implicate moulting fluids in having a major role in the evolution of ant eusociality.

Colonies of eusocial insects are complex biological systems that consist of obligately interacting individuals working cooperatively towards common goals. Colony function and fitness are highly dependent on the types and frequency of interactions between its members, including adults and the various immature stages[5–7]. In ants, for example, adults groom, transport and feed the larvae[2,8]. One important type of social interaction is trophallaxis, the sharing of liquid food in the form of 'social fluids' either mouth-to-mouth or anus-to-mouth. Trophallaxis is widespread among eusocial insects and it is common in many ant species, both between adults and between adults and larvae[5,8–17]. William M. Wheeler, who coined the term, even suggested that trophallaxis would be the key to understanding eusocial behaviour in insects[9,10].

However, because different developmental stages co-occur in close contact inside the nest, different types of social interactions might be confounded or difficult to discern. Therefore, we cannot fully describe the principles of interactions between individuals and how each developmental stage contributes to and benefits from the social fabric of the colony using observations of intact colonies alone. In their 1990 monograph *The Ants*, Bert Hölldobler and Edward O. Wilson noted that this subject "remains largely unexplored", but that "We can reasonably anticipate many surprising new discoveries, some of which may force changes in our thinking about colony organization"[8].

We thus took an approach inspired by reverse engineering to discover new social interactions by isolating individuals, identifying resulting phenotypes, experimentally rescuing those phenotypes, and finally verifying and studying the relevant phenomena in the colony context. We focused specifically on ant pupae, because pupae have been regarded as passive members of the colony and—to our knowledge—no direct contributions of pupae to colony function have been described. We used the clonal raider ant *Ooceraea biroi* as our main model system, before generalizing our findings to other ant species.

## A pupal social fluid hinders survival

To investigate whether and how ant pupae interact with other colony members, we developed a protocol to rear *O. biroi* pupae outside of the colony in social isolation by providing the abiotic requirements for development (Fig. 1a). We isolated pupae from the beginning of pupation until eclosion as adults, and found that they secrete a substantial amount of fluid towards the end of metamorphosis (Fig. 1a,b). Secretion begins six days before eclosion, shortly after the pupae have melanized (Fig. 1a). The secretion initially accumulates in the exuvial space and then exudes from the rectal invagination of the pupal case (Extended Data Fig. 1), forming a stereotypical droplet on the abdominal tip (Fig. 1b). Secretion continues until eclosion, and droplets are often still present on the exuvia. In subsequent experiments, we therefore isolated pupae nine days before eclosion and just before melanization. During this period, on average 99.93 ± 0.22% (mean ± s.d.) of the pupae secrete fluid ($n = 9$, 142–166 pupae per replicate). As long as we manually removed fluid daily, our social isolation protocol yielded high survival rates, with 90 ± 3% of pupae eclosing ($n = 9$, 142–166 pupae per replicate). Among the pupae that survived to eclosion, we observed a

[1]Laboratory of Social Evolution and Behavior, The Rockefeller University, New York, NY, USA. [2]Proteomics Resource Center, The Rockefeller University, New York, NY, USA. [3]Electron Microscopy Resource Center, The Rockefeller University, New York, NY, USA. [4]Howard Hughes Medical Institute, New York, NY, USA. [5]Bioinformatics Resource Center, The Rockefeller University Hospital, The Rockefeller University, New York, NY, USA. [6]Department of Biostatistics, The Rockefeller University Hospital, The Rockefeller University, New York, NY, USA. ✉e-mail: osnir@rockefeller.edu; dkronauer@rockefeller.edu

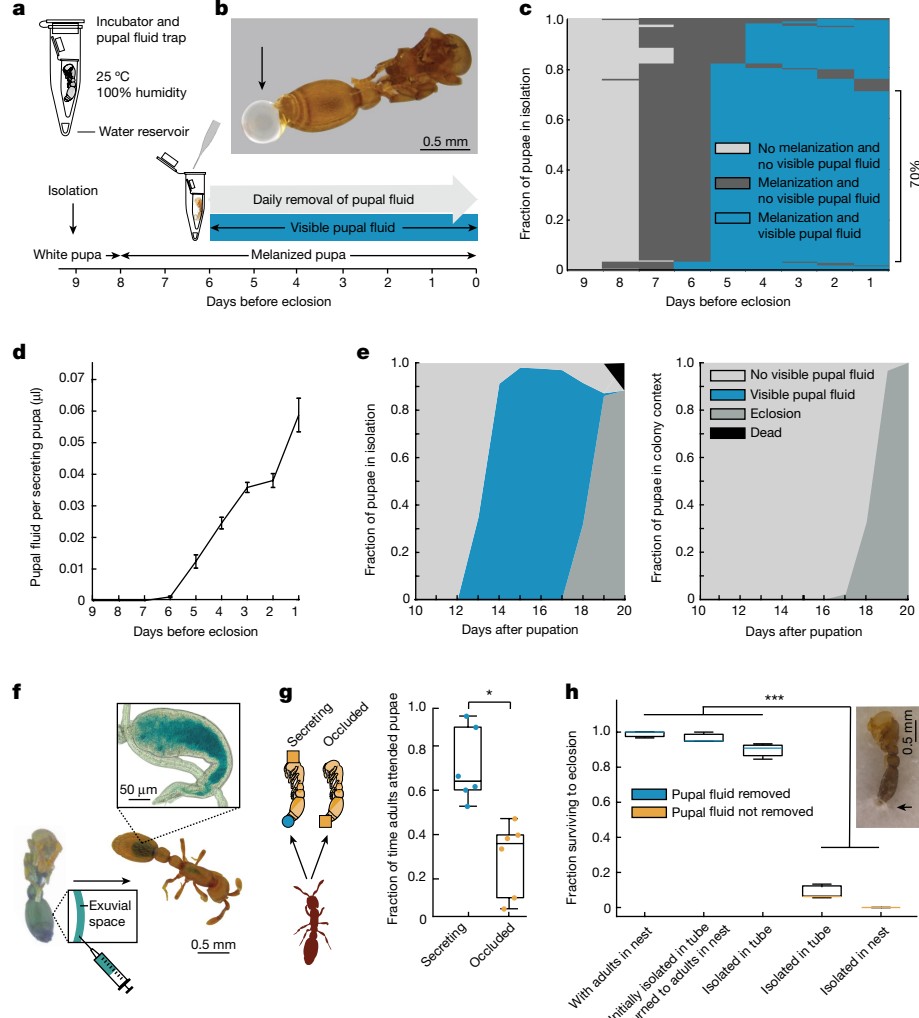

**Fig. 1 | Ant pupae secrete a social fluid that elicits parental care behaviour that is crucial for pupal survival. a**, Developmental timeline and protocol for rearing pupae in social isolation (Methods). **b**, Pupa with secretion droplet accumulated over 24 h in isolation. **c**,**d**, Secretion dynamics of isolated pupae over nine days before eclosion. **c**, Patterns of melanization and secretion during pupal development. *n* = 211 pupae. **d**, The daily social fluid volume per secreting pupa increases over time (Methods). Pearson correlation coefficient (*r*) = 0.93, two-sided *t*-test, $t_{(34)}$ = 14.63, *P* = 3.11 × 10⁻¹⁶. Data are mean ± s.e.m. from *n* = 6 biological replicates with 149–166 pupae each. **e**, Dynamics of pupal secretion and eclosion under social isolation (left; *n* = 942 pupae) and in colonies with adults (right; *n* = 907 pupae). Data are pooled from six biological replicates. **f**, Food dye injected into the pupal exuvial space is taken up by adult ants. The inset shows the dye-filled crop of an adult ant.

**g**, When given the choice, adults spend more time attending to secreting pupae (blue) compared with pupae prevented from secreting (orange) by occluding the tip of the gaster with a paint dot (orange square) (Methods). Secreting pupae received a control dot of paint on the head. One-sided *t*-test, $t_{(5)}$ = −3.04, **P* = 0.014. *n* = 6, with 3 adults, 5 secreting and 5 occluded pupae each. **h**, The survival of pupae to eclosion with (blue) and without (orange) removal of the secretion. One-way ANCOVA with White adjustment. Secretion removal: $F_{(1,16)}$ = 616.67, ****P* = 3.32 × 10⁻¹⁴. *n* = 9 biological replicates for 'isolated in tube' with 'pupal fluid removed', and *n* = 3 for all other conditions (30–166 pupae per replicate). See Supplementary Data 1 for pairwise comparisons. Inset shows a representative pupa isolated in an empty nest, with fungal growth at the time of expected eclosion. Box plots show median (centre line), interquartile range (IQR) (box limits) and 1.5× IQR (whiskers).

stereotypical pattern: Melanization always preceded secretion, 70% of pupae secreted fluid continuously over the course of five days before eclosion (Fig. 1c), and the average daily volume secreted increased over time, up until eclosion (Fig. 1d). Throughout the period of secretion, we collected 23.2 ± 1.7 µl of fluid per biological replicate (*n* = 6, 149–166 pupae per replicate, approximately corresponding to the population of a natural colony of this species).

This large amount of fluid secreted by pupae in isolation and the high fraction of pupae secreting daily (Fig. 1e, left) contrasts with the fact that the fluid cannot be readily observed when pupae are kept inside the colony with adult ants (Fig. 1e, right). We therefore tested whether pupae secrete the fluid in the natural colony setting, and if so, what happens to it. We injected blue food dye into the exuvial space of melanized pupae and traced its distribution in the colony for several

days (*n* = 10 replicate colonies of 10 adults and 10 pupae each) (Fig. 1f). We found blue staining in the digestive system of all adult ants that had spent 24 h with injected pupae (Fig. 1f and Extended Data Fig. 1f). By the end of the experiment, no pupae had died, and all of them responded to tactile stimulation, showing that adults did not acquire the dye by cannibalizing pupae. Instead, this suggests that adults consume the pupal fluid immediately as it is secreted, which is why the fluid does not accumulate and become visible in the colony context. Indeed, adult ants pile up the pupae and remain in prolonged physical contact with them, frequently touching the pupae with their mouthparts (Supplementary Video 1). We then isolated pupae and allowed fluid to accumulate before adding adults. The adults were highly attracted to the fluid and rapidly consumed it (Supplementary Video 2). Preventing extrusion of the secretion by occluding the gaster tip of pupae

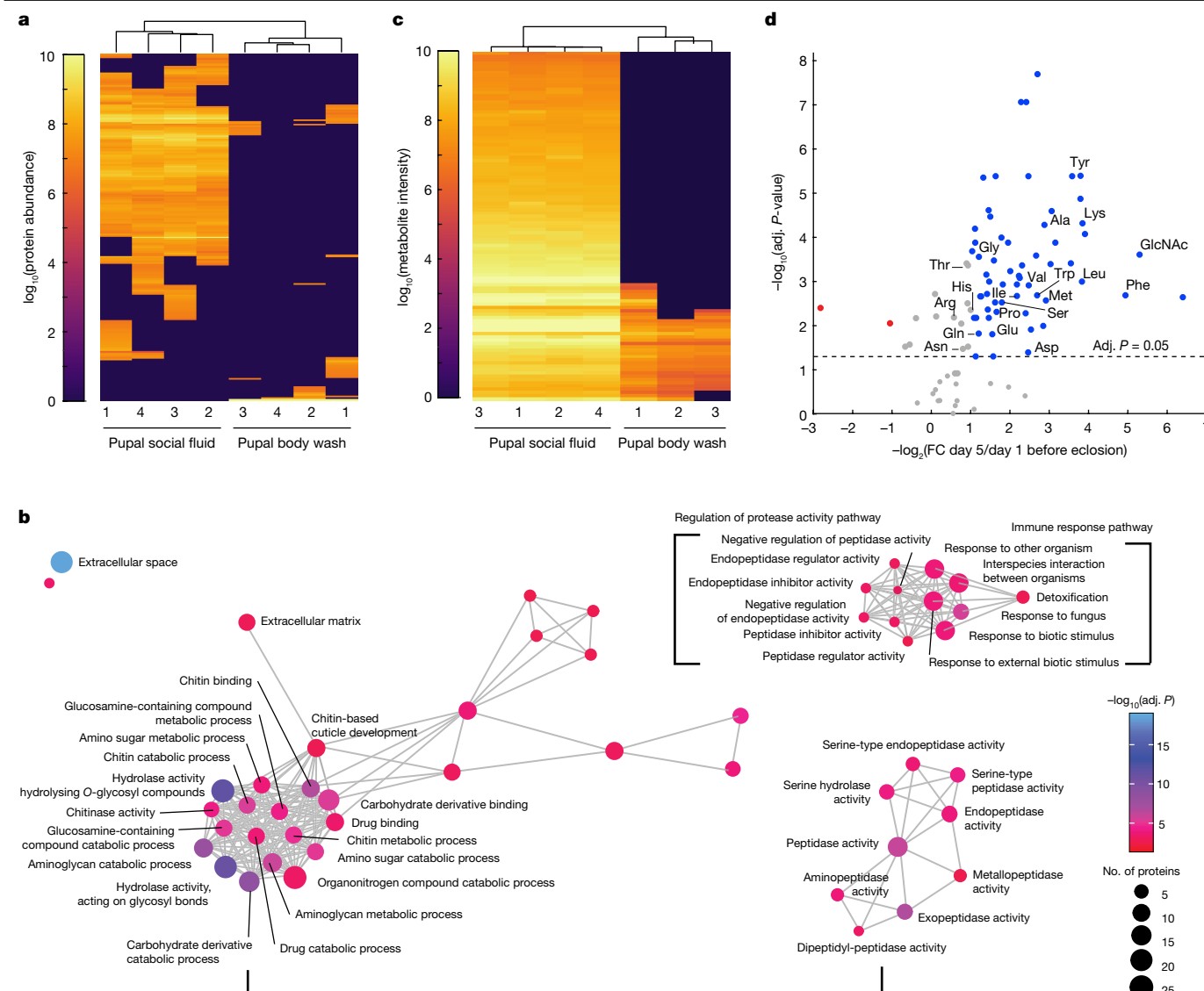

**Fig. 2 | Molecular profiling of the pupal social fluid identifies signatures of insect moulting fluids. a**, Proteomic profiles of pupal social fluid and whole-body wash. Data are from *n* = 4 biological replicates with 426–493 pupae each. Samples collected on days 1–5 before eclosion were pooled. Phylogenies above columns represent similarities between samples. See Extended Data Fig. 2 for details. **b**, Proteomic GO enrichment analysis of pupal social fluid (one-sided hypergeometric test, FDR = 0.05). Nodes correspond to enriched GO terms; node sizes indicate protein numbers in enriched GO terms, and edges connect overlapping protein sets. See Extended Data Fig. 3 for details. Adj. *P*, adjusted *P*-value. **c**, Same as **a**, for metabolomic profiles. See Extended

Data Fig. 4 for details. **d**, Metabolite intensity of time-series samples (days 1–5 before eclosion) normalized to volume secreted per day per pupa from *n* = 4 biological replicates with 426–493 pupae each. Adjusted *P*-values obtained using one-way repeated-measures ANOVA with auto-scaling normalization, FDR = 0.05. Fold change values are between replicate averages from days 5 and 1 before eclosion. Metabolites with |fold change| >2 and significant changes in intensity across time are shown in red (decrease) or blue (increase). Metabolites with no significant change or |fold change| <2 are shown in grey. Points representing free amino acids and GlcNAc are labelled. See Extended Data Fig. 5 for details.

reduced adult attendance compared to secreting controls (Fig. 1g and Supplementary Video 3).

We then tested whether removal of the secretion affects the fitness of pupae. Daily manual removal of the secretion was sufficient to achieve high rates of pupal survival and eclosion in isolation, similar to the colony context in which pupae were in a nest with a group of adults (Fig. 1h). By contrast, if we did not remove fluid from isolated pupae under clean rearing conditions, they drowned in their own secretion (Fig. 1h). However, ant nests are not clean environments. When pupae were isolated in vacant, used nest boxes, the fluid droplets became contaminated with fungi, and these infections spread and ultimately killed all pupae (Fig. 1h). Pupae that were isolated until fluid droplets appeared and then returned to the nest with a group of adults showed high survival

(Fig. 1h). This shows that, in the colony context, pupae depend on adults to remove the secretion, and would otherwise die of fungal infections. Ant pupae are typically considered to be inactive members of the colony, but our experiments show that in *O. biroi*, pupae produce a type of social fluid that is consumed by the adults. This fluid is secreted in large volumes by all pupae during a specific window of development, and elicits parental care behaviour that is necessary for pupal survival.

## The social fluid is derived from moulting fluid

To understand the molecular composition of the pupal social fluid, we collected samples from a population of pupae reared in isolation and conducted proteomic and metabolomic profiling. To verify that the

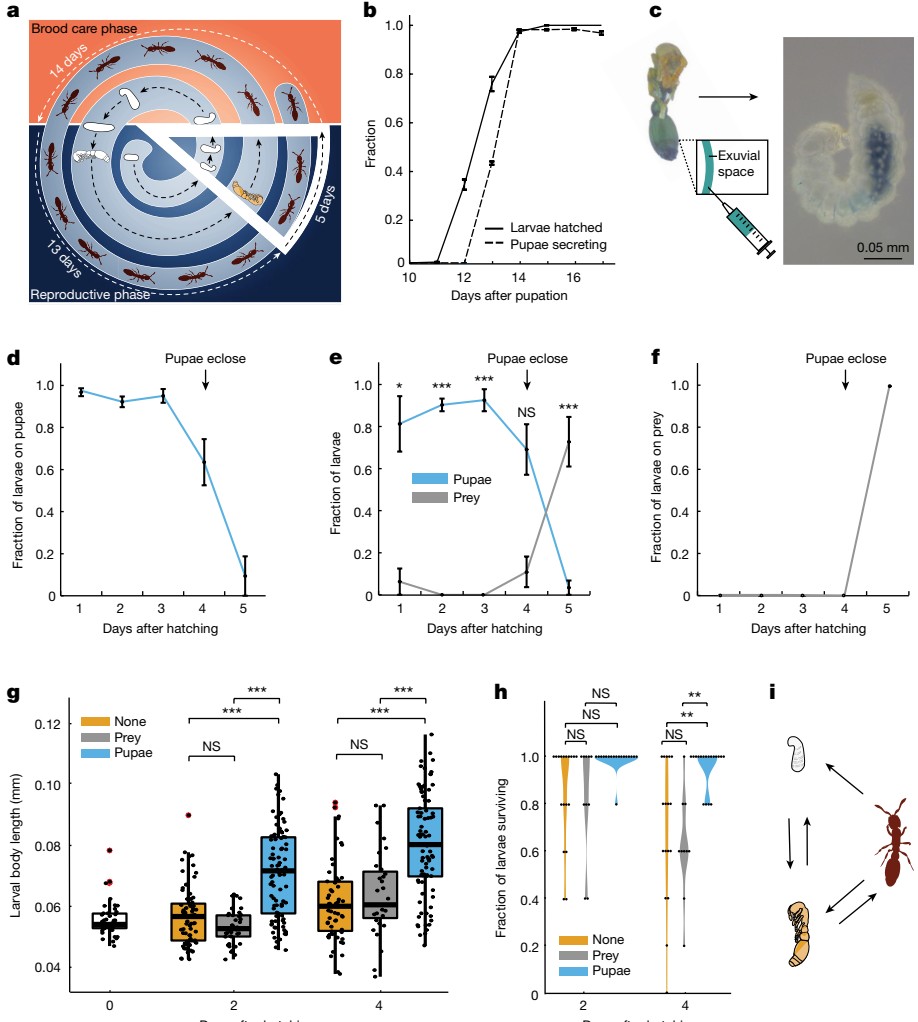

**Fig. 3 | The pupal social fluid serves as a milk-like substance for young larvae. a**, The *O. biroi* colony cycle. The slice (white outline) represents overlap between larvae and pupae of two subsequent cohorts. **b**, Synchronization between larvae hatching (*n* = 4 replicates with 100 larvae each) and pupae secreting (*n* = 3 replicates with 162–166 pupae each). **c**, Larvae consume pupal social fluid. Representative image of 3-day old larva with blue food dye inside the gut after having spent 24 h with dye-injected pupae. **d**–**f**, The preference of adults to place larvae on different food sources. **d**, Pupae only. **e**, Pupae and prey. **f**, Prey only. *n* = 8 colonies per treatment. Nonparametric repeated-measures ANOVA on aligned rank-transformed values followed by two-sided contrasts and FDR = 0.05. **e**, Left to right: $P = 0.02$, $P = 3.53 \times 10^{-4}$, $P = 8.22 \times 10^{-6}$, $P = 0.978$ and $P = 2.65 \times 10^{-12}$. **g**,**h**, Larval growth (**g**) and survival (**h**) during four days after hatching in colonies with only adults (orange), with adults and prey (grey) or with adults and pupae (blue). *n* = 18 colonies with pupae, *n* = 10 with

prey and *n* = 18 with neither. Two-way ANOVA on Box–Cox transformed values with White adjustment followed by two-sided Games–Howell post hoc tests adjusted using Tukey's method. **g**, Larval body length is shown and analysed for individual larvae; 0- to 12-hour-old larvae at the beginning of the experiment (0 days after hatching) are shown in white. Left to right: $P = 0.226$, $P = 1.47 \times 10^{-10}$, $P = 1.55 \times 10^{-15}$, $P = 0.948$, $P = 3.23 \times 10^{-10}$ and $P = 3.45 \times 10^{-4}$. **h**, Left to right: $P = 1$, $P = 0.257$, $P = 0.075$, $P = 0.964$, $P = 0.004$ and $P = 0.002$. **i**, Contributor-to-beneficiary interactions between pupae, young larvae and adults involving the pupal social fluid. Pupae secrete their moulting fluid and adults place larvae on pupae. Larvae and adults consume the fluid, and fluid consumption prevents pupal infections and death. In **b**,**d**–**f**, data are mean ± s.e.m. For pairwise comparisons in **e**,**g**,**h**, see Supplementary Data 4–6. Box plots are as defined in Fig. 1. Outliers are shown in red.

detected compounds were secreted with the fluid, rather than stemming from contamination acquired via contact with the surface of the pupae, we compared the pupal fluid profiles with profiles obtained from pupal whole-body washes. We collected samples daily during the five days before pupae eclosed, and analysed them using liquid chromatography–tandem mass spectrometry (LC–MS/MS). Hierarchical clustering of proteomic profiles revealed a clear distinction between the pupal fluid and pupal whole-body wash (Fig. 2a and Extended Data Fig. 2). Out of 212 identified proteins, 185 were either found exclusively or significantly enriched in pupal fluid compared with pupal whole-body wash (*t*-tests, false discovery rate (FDR) = 0.05) (Supplementary Data 2). Proteomic Gene Ontology (GO) enrichment analysis identified the processes of protein and chitin degradation, which are represented

by functions such as peptidase and chitinase activity, respectively (Fig. 2b and Extended Data Fig. 3). Activities related to protein and chitin degradation are involved in the degradation of the old cuticle, which is one of three major molecular pathways characteristic of insect moulting fluids, as has been demonstrated by proteomic profiling and functional analyses of *Manduca sexta*, *Bombyx mori* and *Tribolium castaneum* moulting fluids[18–21]. The other two major pathways found in moulting fluids are immune response and the regulation of protease activity[18–21]. GO terms associated with both pathways are enriched in the pupal fluid (Fig. 2b). The immune response pathway includes enzymes involved in cuticle melanization, such as phenoloxidase[20–23], consistent with our observation that the onset of melanization always precedes pupal secretion (Fig. 1c).

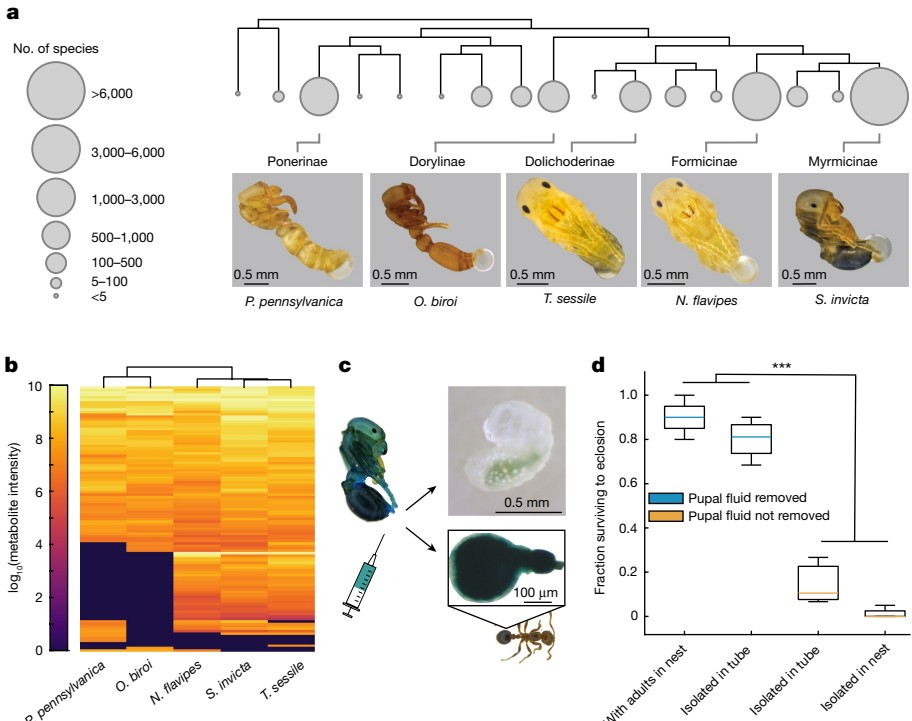

**Fig. 4 | The social function of the pupal moulting fluid is conserved.**
**a**, Subfamily-level ant phylogeny (top) and pupae of representative species with secretion droplets accumulated over 24 h in social isolation. In *P. pennsylvanica*, the cocoon has been removed to better show the secretion (also see Extended Data Fig. 8). At least 30 pupae were isolated for each species, with consistent results. **b**, Metabolomic profiles of pupal social fluid of the species in **a**. Data are from 1 replicate with 30 pupae each. Samples collected on days 1–4 before eclosion were pooled. The phylogeny above columns represents similarities between species. See Extended Data Fig. 7 for details. **c**, *S. invicta* adults and early instar larvae take up food dye injected into the exuvial space of pupae. The inset shows the stomach of an adult ant. $n = 10$ replicates of 10 adults, 10 pupae and 5 early instar larvae each. **d**, Survival of pupae to eclosion with (blue) and without (orange) removal of the secretion under the following conditions: with adults in nest (unclean conditions), isolated in tube (isolated in clean conditions), and isolated in nest (isolated in a vacant nest in unclean conditions). Two-way ANOVA with White adjustment for heteroskedasticity. Secretion removal: $F_{(1,11)} = 51.38$, ***$P = 1.83 \times 10^{-5}$. $n = 3$ biological replicates for 'isolated in tube' with 'pupal fluid not removed' and $n = 4$ for all other conditions (30 pupae per replicate). Box plots are as defined in Fig. 1. See Supplementary Data 7 for pairwise comparisons.

Moulting fluids contain proteolytic and chitinolytic enzymes that catalyse the breakdown of chitin into free amino acids and *N*-acetyl-D-glucosamine[19] (GlcNAc). To provide additional evidence that the pupal fluid is derived from the moulting fluid, we screened for these breakdown products. Targeted metabolomic profiling of pupal fluid and pupal whole-body wash indicated that the social fluid contains a variety of micro- and macronutrients. These include all essential amino acids, multiple carbohydrates including GlcNAc, nucleic acids and vitamins (Fig. 2c, Extended Data Fig. 4). Out of 107 metabolites identified, 105 were significantly enriched in the pupal fluid compared to pupal whole-body wash (*t*-tests, FDR = 0.05) (Supplementary Data 3). Time-series analysis revealed an increase in many free amino acids and GlcNAc as pupae approached eclosion. Moreover, GlcNAc was among the metabolites with the largest fold change (Fig. 2d). This is consistent with findings from other insects, where these metabolites increase in the moulting fluid during the moulting process[19,24]. These results show that the pupal social fluid is rich in a variety of proteins and metabolites and has the molecular and physiological characteristics of insect moulting fluids. Non-eusocial Ecdysozoa reabsorb the moulting fluid before ecdysis to recycle nutrients[25,26]. By contrast, clonal raider ant pupae secrete a large proportion of the moulting fluid from the pupal case, where it is consumed by adult nestmates.

## Young larvae consume the social fluid

Given the nutritive content of the pupal social fluid, we hypothesized that larvae, which grow and therefore require the most food, might also consume it. Colonies of *O. biroi* undergo phasic cycles that alternate between brood care and reproductive phases (Fig. 3a). During the brood care phase, colonies contain late instar larvae that induce workers to leave the nest and forage for food. During the reproductive phase, colonies contain pupae, foraging activity ceases, and workers synchronously lay a new batch of eggs that hatch into larvae at the end of the reproductive phase (Fig. 3a). Brood therefore develops in discrete cohorts, and each new cohort of larvae hatches while the previous cohort is at the late pupal stage[6,27,28]. Thus, the pupal fluid might provide an important food source for larvae before workers begin foraging (approximately the first 4 days of their lives) (Fig. 3a). To test whether larvae are present during the time in the colony cycle when pupae secrete fluid, we isolated pupae and eggs from colonies in the middle of the reproductive phase and daily recorded the number of pupae secreting and the number of larvae hatching. The first larvae hatched only one day before pupae began secreting, showing that the two events are tightly synchronized (Fig. 3b). We then placed larvae in a colony with pupae injected with food dye to test whether the larvae consume dye-stained pupal fluid. We consistently observed dye staining in larval guts within 24 h of the experiment (Fig. 3c). In the colony context, young larvae are usually attached to pupae, often with their mouthparts, suggesting that they consume the liquid directly as it appears on the pupal surface (Supplementary Video 4).

Larvae have limited mobility and are thus dependent on adults to place them on food sources. Observations suggest that *O. biroi* adults readily place young larvae on pupae and older larvae on prey items (Supplementary Video 5). To test the adults' preference for alternative

food sources for young larvae, we provided colonies of ten adults and ten newly hatched larvae with either ten melanized *O. biroi* pupae, both ten melanized *O. biroi* pupae and ten prey items (dead pupae of the fire ant *Solenopsis invicta*, replaced every other day), or ten prey items alone. We then recorded the location of larvae once a day for five days. When given pupae alone, ants placed almost all larvae on pupae until eclosion (Fig. 3d). When given a choice, ants showed a strong preference for placing the larvae on pupae as opposed to prey (Fig. 3e and Extended Data Fig. 6a). Even in the absence of pupae, ants did not place young larvae on prey (Fig. 3f and Extended Data Fig. 6b). Thus, ants place newly hatched and early instar larvae on their own pupae, where they have the ability to feed on pupal fluid, and only place later instar larvae on prey (Fig. 3d–f and Extended Data Fig. 6). To determine whether this relationship contributes to larval growth and survival, we conducted an experiment in which larvae were hatched in social isolation and, at 0–12 h of age and without prior access to food, were fostered into colonies of adult ants. Colonies were composed of ten adults and five fostered larvae each, and were supplied with either ten melanized *O. biroi* pupae, ten prey items (dead *S. invicta* pupae, replaced every other day), or neither. We measured larval growth and survival over four days after hatching. Larvae in colonies with pupae grew significantly more (Fig. 3g) and had significantly higher survival (Fig. 3h) compared with larvae in colonies without pupae, regardless of whether prey was present or not. All pupae in these colonies eclosed, showing that pupae themselves were not consumed (brood cannibalism is a commonly observed response to stressful conditions in ant colonies). There were no significant differences in growth and survival between larvae in colonies that were given prey and those kept without prey (Fig. 3g,h). Thus, we observed faster growth and higher survival of larvae reared in the presence of pupae, and found no evidence for a similar beneficial effect of prey items. These experiments show that the pupal fluid serves as a 'milk-like' substance for newly hatched larvae, greatly increasing larval growth and survival during the first days after hatching. Together, our results reveal that the pupal moulting fluid has a previously unknown yet important role in *O. biroi* social organization and colony fitness (Fig. 3i).

## Pupal secretions are conserved across the ants

Despite its importance in *Ooceraea biroi* (ant subfamily Dorylinae)—to our knowledge—a social function of pupal secretions as trophallactic fluids has not been described in any of the > 14,000 ant species. To test whether this function is specific to *O. biroi*, we socially isolated pupae from four other species to cover the five major ant subfamilies: *S. invicta* (subfamily Myrmicinae), *Nylanderia flavipes* (Formicinae), *Tapinoma sessile* (Dolichoderinae) and *Ponera pennsylvanica* (Ponerinae). We found that melanized pupae of all four species secrete fluid droplets from the abdominal tip (Fig. 4a). Metabolomic profiling showed that the composition of these secretions is similar to that of *O. biroi*: out of the 70 metabolites identified in *O. biroi* in this analysis, we found 69 in at least one other ant species, including free amino acids and GlcNAc (Fig. 4b and Extended Data Fig. 7). This shows that, across the ants, pupae secrete a liquid derived from the moulting fluid.

To test whether pupal fluids also have a social role in other ant species, we injected blue food dye into the exuvial space of melanized *S. invicta* pupae and placed them into small colony fragments. After 24 h, the digestive systems of all adults and most early instar larvae were stained blue, indicating that they had ingested pupal fluid (Fig. 4c). A previous study showed that *S. invicta* pupae do not eclose in social isolation, and concluded that adults are required to help remove the exuvia during eclosion[29]. By contrast, while pupae died if their secretion was not removed, manual removal of the social fluid was sufficient to achieve high survival rates without any additional assistance during eclosion (Fig. 4d). Together, our results show that the secretion of moulting fluid by pupae is widely conserved in ants, and that the

dependence of pupae on other colony members to consume the fluid is not limited to *O. biroi*. In many ant species, adults place young larvae on pupae (Supplementary Video 6), and that adults drink dyed pupal fluid is directly visible in *N. flavipes*, because the abdomen of these ants is partly translucent (Supplementary Video 7). In species whose pupae are enclosed in cocoons, such as *P. pennsylvanica*, adults consume the social fluid through the permeable silken fabric (Extended Data Fig. 8). This can be visualized using dyes in ants with translucent cuticles, such as *Myrmecocystus mexicanus* (subfamily Formicinae) (Extended Data Fig. 8 and Supplementary Video 8). Additional work will be required to study the social role of pupal fluids across the ant tree of life.

Although the social functions of the pupal secretion must be a derived trait in ants, it remains to be determined whether pupae of solitary hymenopterans produce secretions derived from the moulting fluid. If so, these secretions might have readily been co-opted during the various independent evolutionary origins of hymenopteran eusociality. We examined this possibility by socially isolating honeybee (*Apis mellifera*) pupae. None of the pupae produced visible secretion droplets (Extended Data Fig. 9), and more than 80% of the pupae survived to eclosion in social isolation without additional assistance. Although this shows that not all eusocial clades rely on the social functions of pupal fluids, the evolutionary dynamics of pupal secretions across the Hymenoptera will be a rewarding avenue for future investigation.

## Discussion

During major evolutionary transitions, initially independent units become integrated and mutually dependent to form higher levels of biological organization. Here we show that ant pupal moulting fluid has acquired novel social functions that create a previously unrecognized interdependence between pupae, larvae and adults. We have demonstrated that this fluid is detrimental for pupae if not removed, and that it is an important food source for early larvae. This secretion is likely to have additional far-reaching effects on larvae and other colony members. Previous work has shown that trophallactic fluid from adult ants contains not only pre-digested food, but also endogenous molecules such as juvenile hormone, a known regulator of insect development and behaviour[15]. Similarly, the pupal fluid described here contains hormones and neuroactive substances that may modulate the development and behaviour of larvae and adults (see Supplementary Discussion for details). Far from being passive colony members, pupae thus have an active and central role in ant colony organization.

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

# Methods

## Rearing *O. biroi* pupae in social isolation and collecting pupal fluid

In *O. biroi* colonies, larvae and pupae develop in discrete and synchronized cohorts[26]. Ten days after the first larvae had entered pupation in a large stock colony, the entire colony was anaesthetized using a $CO_2$ pad, and white pupae were separated using a paintbrush. Pupae were individually placed in 0.2 ml PCR tubes with open lid. These tubes were then placed inside 1.5 ml Eppendorf tubes with 5 µl sterile water at the bottom to provide 100% relative humidity. The outer tubes were closed and kept in a climate room at 25 °C. The inner tube in this design prevents the pupa from drowning in the water reservoir. The outer tubes were kept closed throughout the experiment, except for once a day when the tubes were opened to remove pupal social fluid. Pulled glass capillaries were prepared as described elsewhere[29], and used to remove and/or collect secretion droplets. We were careful to leave no remains of the secretion behind on the pupae or the inside of the tubes. To ensure that all secretion had been removed, pupae were taken out of the tube after fluid collection and briefly placed on a tissue paper to absorb any excess liquid. The inner tubes were replaced if needed – for example, if fluid traces were visible on the old tube after collection. Each pupa was checked daily for secretion (absent or present), onset of melanization and eclosion, and whether the pupa was alive (responding to touch). Control groups of 30 pupae and 30 adult ants from the same stock colony and cohort as the isolated pupae were placed in Petri dishes with a plaster of Paris floor, and the same parameters as for the isolated pupae were scored daily. Experiments ended when all pupae had either eclosed or died. Newly eclosed (callow) workers moved freely inside the tube and showed no abnormalities when put in a colony. A pupa was declared dead if it did not shed its pupal skin and did not respond to touch three days after all pupae in the control group had eclosed.

To calculate the average secretion volume per secreting pupa (Fig. 1d), the total volume collected daily from a group of isolated pupae (142–166 pupae) was divided by the number of pupae from which fluid had been collected that day. The total volume was determined by multiplying the height of the fluid's meniscus in the capillary by $\pi r^2$, where r is the inner radius of the capillary (0.29 mm). While pupae were secreting, pupal whole-body wash samples were collected daily. The pupae were removed from colonies with adults and washed promptly with 1500 µl LC–MS grade water. Whole-body wash samples were lyophilized and reconstituted in 15 µl LC–MS grade water.

## Collecting additional ant species and honeybees, rearing pupae in social isolation, and collecting pupal fluids

Colonies of the ants *N. flavipes*, *T. sessile*, *P. pennsylvanica* and *Lasius neoniger* were collected in NY state, USA (Central Park, Manhattan; Pelham Bay Park, Bronx; Prospect Park, Brooklyn; and Woodstock). *Solenopsis invicta* colonies were collected in Athens, GA, USA. *M. mexicanus* colonies were collected in Piñon Hills, CA, USA. Colonies comprised of queens, workers and brood were maintained in the laboratory in airtight acrylic boxes with plaster of Paris floors. Colonies were fed a diet of insects (flies, crickets and mealworms). White pupae were socially isolated, cocoons were removed in the case of *P. pennsylvanica*, and secretion droplets were collected from melanized pupae as described for *O. biroi*. *A. mellifera* pupae of unknown age were socially isolated from hive fragments (A&Z Apiaries, USA) and reared as described for *O biroi*, except that the rearing temperature was set to 32 °C. Relative humidity was set to either 100% to replicate conditions used for the different ant species, or to 75% as recommended in the literature[30].

## Injecting dye and tracking pupal fluid

Injection needles were prepared as in previous studies[31]. Injections were performed using an Eppendorf Femtojet with a Narishige micromanipulator. The Femtojet was set to Pi 1000 hPa and Pc 60 hPa. Needles were broken by gently touching the capillary tip to the side of a glass slide. To inject, melanized pupae were placed on 'Sticky note' tape (Post-it), with the abdomen tip forward and the ventral side upward. Pupae were injected with blue food colouring (McCormick) into the exuvium for 1–2 s by gently piercing the pupal case at the abdominal tip with the needle. During successful injections, no fluid was discharged from the pupa when the needle was removed, and the moulting fluid inside the exuvium was immediately stained. Pupae were washed in water three times to remove any excess dye. Following injections, 10 pupae were reared in social isolation to confirm the secretion of dyed droplets. For experiments, injected pupae were transferred to colonies with adult ants (Figs. 1f and 4c) or to colonies with adult ants and larvae (Figs. 3b and 4c) to track the distribution of the pupal social fluid.

After spending 24 h with dye-injected pupae, adults were taken out of the colony, briefly immersed in 95% ethanol, and transferred to PBS. Digestive systems were dissected in cold PBS and mounted in DAKO mounting medium. Crop and stomach images (Fig. 1f, inset and Fig. 4c, inset) were acquired with a Revolve microscope (Echo). Larvae are translucent, and the presence of dye in the digestive system can be assayed without dissection. Whole-body images of larvae were acquired with a Leica Z16 APO microscope equipped with a Leica DFC450 camera and Leica Application Suite version 4.12.0 (Leica Microsystems). In the experiment on larval growth (Fig. 3c), larval length was measured from images using ImageJ[32].

## Occluding pupae

Ten pupae were placed on double-sided tape on a glass coverslip with the ventral side up. The area between the pupae was covered with laser-cut filter paper to prevent adults from sticking to the tape. The pupae were then placed in a 5 cm diameter Petri dish with a moist plaster of Paris floor. To block pupal secretion, the tip of the gaster was occluded with a drop of oil-paint (Uni Paint Markers PX-20), which has no discernible toxic effect[7]. Secreting pupae received a drop of the same paint on their head to control for putative differences resulting from the paint. Pupae were left in isolation for one day before adults were added to the assay chamber.

## Behavioural tracking of adult preference assay

Videos were recorded using BFS-U3-50S5C-C: 5.0 MP, 35 FPS, Sony IMX264, Colour cameras (FLIR) and the Motif Video Recording System (Loopbio). To assess adult preference (Fig. 1g), physical contact of adults with pupae was manually annotated for the first 10 min after the first adult had encountered (physically contacted) a pupa.

## Protein profiling

We extracted 30 µl of pupal social fluid and whole-body wash samples with 75:25:0.2 acetonitrile: methanol: formic acid. Extracts were vortexed for 10 min, centrifuged at 16,000g and 4 °C for 10 min, dried in a SpeedVac, and stored at −80 °C until they were analysed by LC–MS/MS.

Protein pellets were dissolved in 8 M urea, 50 mM ammonium bicarbonate, and 10 mM dithiothreitol, and disulfide bonds were reduced for 1 h at room temperature. Alkylation was performed by adding iodoacetamide to a final concentration of 20 mM and incubating for 1 h at room temperature in the dark. Samples were diluted using 50 mM ammonium bicarbonate until the concentration of urea had reached 3.5 M, and proteins were digested with endopeptidase LysC overnight at room temperature. Samples were further diluted to bring the urea concentration to 1.5 M before sequencing-grade modified trypsin was added. Digestion proceeded for 6 h at room temperature before being halted by acidification with TFA and samples were purified using in-house constructed C18 micropurification tips.

LC–MS/MS analysis was performed using a Dionex3000 nanoflow HPLC and a Q-Exactive HF mass spectrometer (both Thermo Scientific). Solvent A was 0.1% formic acid in water and solvent B was 80% acetonitrile, 0.1% formic acid in water. Peptides were separated on a 90-minute

linear gradient at 300 nl min⁻¹ across a 75 µm × 100 mm fused-silica column packed with 3 µm Reprosil C18 material (Dr. Maisch). The mass spectrometer operated in positive ion Top20 DDA mode at resolution 60 k/30 k (MS1/MS2) and AGC targets were $3 \times 10^6/2 \times 10^5$ (MS1/MS2).

Raw files were searched through Proteome Discoverer v.1.4 (Thermo Scientific) and spectra were queried against the *O. biroi* proteome using MASCOT with a 1% FDR applied. Oxidation of M and acetylation of protein N termini were applied as a variable modification and carbamidomethylation of C was applied as a static modification. The average area of the three most abundant peptides for a matched protein[33] was used to gauge protein amounts within and between samples.

### Functional annotation and gene ontology enrichment

To supplement the current functional annotation of the *O. biroi* genome[34], the full proteome for canonical transcripts was retrieved from UniProtKB (UniProt release 2020_04) in FASTA format. We then applied the EggNog-Mapper tool[35,36] (http://eggnog-mapper.embl. de, emapper version 1.0.3-35-g63c274b, EggNogDB version 2) using standard parameters (m diamond -d none --tax_scope auto --go_evidence non-electronic --target_orthologs all --seed_ortholog_evalue 0.001 --seed_ortholog_score 60 --query-cover 20 --subject-cover 0) to produce an expanded annotation for all GO trees (Molecular Function, Biological Process, Cellular Components). The list of proteins identified in the pupal fluid was evaluated for functional enrichment in these GO terms, *P*-values were adjusted with an FDR cut-off of 0.05, and the network plots were visualized using the clusterProfiler package[37].

### Metabolite profiling

For bulk polar metabolite profiling, we used 10 µl aliquots of pupal social fluid and whole-body wash (pooled samples). For the time-series metabolite profiling, 1 µl of pupal social fluid and whole-body wash was used. Samples were extracted in 180 µl cold LC−MS grade methanol containing 1 µM of uniformly labelled ¹⁵N- and ¹³C-amino acid internal standards (MSK-A2-1.2, Cambridge Isotope Laboratories) and consecutive addition of 390 µl LC−MS grade chloroform followed by 120 µl of LC−MS grade water.

The samples were vortexed vigorously for 10 min followed by centrifugation (10 min at 16,000g and 4 °C). The upper polar metabolite-containing layer was collected, flash frozen and SpeedVac-dried. Dried extracts were stored at −80 °C until LC−MS analysis.

LC−MS was conducted on a Q-Exactive benchtop Orbitrap mass spectrometer equipped with an Ion Max source and a HESI II probe, which was coupled to a Vanquish UPLC system (Thermo Fisher Scientific). External mass calibration was performed using the standard calibration mixture every three days.

Dried polar samples were resuspended in 60 µl 50% acetonitrile, and 5 µl were injected into a ZIC-pHILIC 150 × 2.1 mm (5 µm particle size) column (EMD Millipore). Chromatographic separation was achieved using the following conditions: buffer A was 20 mM ammonium carbonate, 0.1% (v/v) ammonium hydroxide (adjusted to pH 9.3); buffer B was acetonitrile. The column oven and autosampler tray were held at 40 °C and 4 °C, respectively. The chromatographic gradient was run at a flow rate of 0.150 ml min⁻¹ as follows: 0−22 min: linear gradient from 90% to 40% B; 22−24 min: held at 40% B; 24−24.1 min: returned to 90% B; 24.1−30 min: held at 90% B. The mass spectrometer was operated in full-scan, polarity switching mode with the spray voltage set to 3.0 kV, the heated capillary held at 275 °C, and the HESI probe held at 250 °C. The sheath gas flow was set to 40 units, the auxiliary gas flow was set to 15 units. The MS data acquisition was performed in a range of 55−825 *m/z*, with the resolution set at 70,000, the AGC target at $10 \times 10^6$, and the maximum injection time at 80 ms. Relative quantification of metabolite abundances was performed using Skyline Daily v 20.1 (MacCoss Lab) with a 2 ppm mass tolerance and a pooled library of metabolite standards to confirm metabolite identity (via

data-dependent acquisition). Metabolite levels were normalized by the mean signal of 8 heavy ¹³C,¹⁵N-labelled amino acid internal standards (technical normalization).

The raw data were searched for a targeted list of ~230 polar metabolites and the corresponding peaks were integrated manually using Skyline Daily software. We were able to assign peaks to 107 compounds based on high mass accuracy (<2 ppm mass deviation) and retention time accuracy (<12 s deviation from known standards). A pool of all the biological samples was used for quality control and analysed using a data-dependent Top2 MS/MS scan (with polarity switching) to acquire MS/MS data and further validate metabolite identity. The data-dependent MS/MS scans were acquired at a resolution of 17,500, $1 \times 10^5$ AGC target, 50 ms max injection time, 1.6 Da isolation width, stepwise normalized collision energy (NCE) of 20, 30, 40 units, 8 s dynamic exclusion, and loop count of 2.

### Scanning electron microscopy

All sample preparation steps for scanning electron microscopy were performed under a fume hood at room temperature, unless indicated otherwise. Scintillating vials containing *O. biroi* pupae were placed on a rotator to ensure uniform treatment. One- and fifteen-days old pupae (*n* = 9 each) were incubated overnight at 4 °C in a fixative solution containing 2% glutaraldehyde, 2% paraformaldehyde in 0.1 M sodium cacodylate buffer, pH 7.2. Pupae were then washed 4 times with 0.1 M sodium cacodylate, pH 7.2, for 5 min each, followed by incubation for 2.5 h at 4 °C in a 1% osmium tetroxide solution in 0.1 M sodium cacodylate buffer, pH 7.2. Subsequently, pupae were washed four times with Milli-Q purified water. A graded series of ethanol was used to dehydrate the washed samples. The dehydration steps until 70% ethanol were performed at 4 °C, and thereafter the samples were dehydrated (90% and 100% ethanol) at room temperature. After complete dehydration, the pupae were dried with hexamethyldisilazane as described previously[38]. Afterwards, samples were mounted on a carbon tape placed on a flat stub and coated with a 12 nm thick layer of iridium nanoparticles using a Leica EM ACE600 sputter coater. The coated samples were imaged with a Jeol JSM-IT500HR scanning electron microscope. Cocoons of *P. pennsylvanica* were directly mounted on a carbon tape, coated, and imaged as described above for *O. biroi* pupae.

### Instant structured illumination microscopy

For imaging, 13- to 17-day-old melanized pupae were placed in the microwell of a 35 mm glass bottom Petri dish (MatTek) and covered with a coverslip. Abiotic conditions (100% humidity, 25 °C) were maintained throughout the recording using an environmental stage chamber (Okolab). Bright-field images of pupal fluid droplets secreted from the abdominal tip were acquired with a DMi8 inverted microscope (Leica), an Orca fusion CMOS camera (Hamamatsu) and the VisiTech InstantSIM (iSIM) (VisiTech International) real-time superresolution system with a 20×/0.75 water objective at a resolution of 3.08 pixels per µm. Imaging was performed at different depths. The z-plane depths were selected to optimize visualization of secretion droplets, the rectal invagination, and the genital opening. Images were acquired using VisiView acquisition software version 4.5.0.13 (VisiTech International) and image processing was performed using FIJI/ImageJ version 1.52p[32].

### Statistics and reproducibility

Statistical analyses were performed in MATLAB R2020b (MathWorks) and R version 3.6.3. In datasets used for ANOVA or ANCOVA, normality was assessed by examining Q−Q plots of model residuals and Box−Cox transformation was applied when needed. For heteroskedasticity, White adjustment was applied when appropriate. ANOVA and ANCOVA models were followed by two-sided Games−Howell post hoc tests for multiple comparisons and the *P*-values obtained were adjusted for multiple testing using Tukey's method. A nonparametric repeated-measures ANOVA model with aligned rank transformation was used to analyse

the larvae placement data. The model included treatment, day, and treatment-by-day interaction. Estimated marginal means were computed and used for two-sided post hoc pairwise comparisons with FDR = 0.05. The time-series metabolomic dataset was analysed with MetaboAnalyst version 4.0[39] using one-way repeated-measures ANOVA with auto-scaling normalization and FDR = 0.05. Details of statistical tests are given in the respective figure legends.

Figure 1b shows a representative pupa with a secretion droplet out of $n = 1,368$ pupae observed under identical conditions. Figure 1f shows a representative pupa and the crop of a representative adult out of a total of $n = 10$ replicate colonies of 10 adults and 10 pupae each. The inset of Fig. 1h shows a representative pupa out of a total of $n = 90$ pupae.

Figure 3c shows a representative larva that has ingested dyed pupal secretion out of $n = 3$ replicate colonies with 10 larvae, adults and pupae each.

Experimental ants were collected haphazardly from stock colonies and distributed across experimental colonies or setups at random. The order of collection and place in which the colonies or setups were arranged were randomized. Blinding was not applicable in this study due to the descriptive nature of the experiments. Sample size calculations were not performed for the experiments. Appropriate sample sizes were estimated based on preliminary data that showed strong effects with small variations. The study included one- to several-month-old females, as well as females of different developmental stages (larvae and pupae). Only female ants display social behaviour, and all worker ants are female. In *O. biroi* in particular, males are only produced sporadically and do not partake in the social life of the colony. Conducting experiments with males is therefore not relevant in the context of the current study.

### Reporting summary

Further information on research design is available in the Nature Portfolio Reporting Summary linked to this article.

### Data availability

Mass spectrometry proteomics data have been uploaded to the PRIDE database (accession number PXD037344). Source data are provided with this paper.

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

**Acknowledgements** We thank P. Piekarski for help with identifying ants; L. Olivos-Cisneros for help with ant maintenance; H. Zeng and K. Ross for supplying *S. invicta* colonies; A. Fuller for supplying *A. mellifera* samples; I. Lacroix for collecting ants; C. Pyrgaki for microscopy related advice; A. Gal, J. Petrillo and P. Strogies for help in designing and building the tracking setup; and K. Lacy, V. Chandra and W. Trible for feedback on the manuscript. This work was supported by the National Institute of General Medical Sciences of the NIH under Award R35GM127007 to D.J.C.K. The content is solely the responsibility of the authors and does not necessarily represent the official views of the NIH. Additional support was provided by a Faculty Scholars Award from the Howard Hughes Medical Institute to D.J.C.K., National Center for Advancing Translational Sciences of the NIH grant UL1 TR001866 to C.S.J., and a Gruss Lipper Post-Doctoral Research Fellowship and a Leon Levy Neuroscience Fellowship to O.S. D.J.C.K. is an investigator of the Howard Hughes Medical Institute. This is Clonal Raider Ant Project paper number 20.

**Author contributions** O.S. and D.J.C.K. conceived and designed experiments. O.S. performed experiments and analysed and visualized the data. H.A. and S.H. performed metabolomics and proteomics analyses, respectively. A.S. performed scanning electron microscopy imaging. T.S.C. performed enrichment analyses for Fig. 2b. O.S. and C.S.J. performed statistical analyses. J.R. and O.S. performed iSIM imaging. O.S., S.V.-R. and D.J.C.K. conducted fieldwork. O.S. and D.J.C.K. wrote the manuscript. D.J.C.K. supervised the project.

**Competing interests** The authors declare no competing interests.

**Additional information**
**Correspondence and requests for materials** should be addressed to Orli Snir or Daniel J. C. Kronauer.

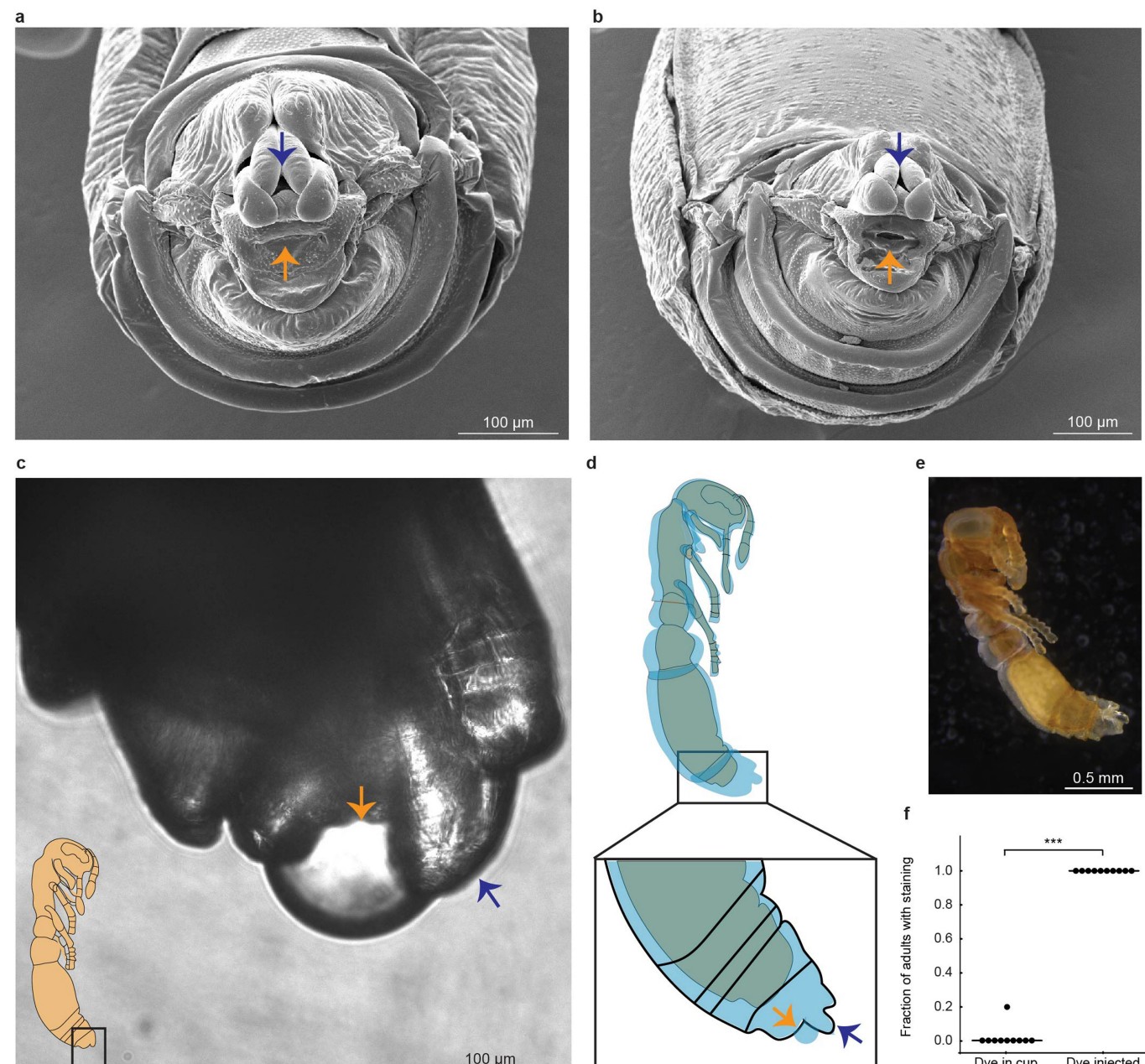

**Extended Data Fig. 1 | The pupal secretion exudes from the rectal invagination of the pupal case.** (a-b) Scanning electron microscopy images of young (1 day old) (a) and old/melanized (15 days old) (b) *O. biroi* pupae. Arrows indicate the genital opening (blue) and the rectal invagination (orange). The rectal invagination is closed in young pupae (a) and only forms a clear opening in older pupae (b). (a-b) show representative images out of n = 9 examined specimens each that yielded consistent results. (c) Instant structured illumination microscopy image of the abdominal tip of a melanized *O. biroi* pupa showing a droplet of pupal fluid secreted from the rectal invagination (orange arrow). No secretion is observed above the genital opening (blue arrow). This experiment was repeated with n = 3 pupae, yielding consistent results, and one representative image is shown here. (d) Schematic illustrating the pupal fluid (blue) in the exuvial space surrounding the pupa (top). The pupal fluid exudes from the rectal invagination of the pupal case (orange arrow, bottom). (e) Air injected into the pupal exuvial space shortly before eclosion visualizes the approximate distribution and volume of this space around the developing pupa. It also shows that the genital opening and rectal invagination both directly connect the exuvial space to the outside, without specialized ducts. Consistent results were obtained from n = 3 pupae, and one representative image is shown here. (f) Adult *O. biroi* show staining of the digestive system only after spending 24 h with pupae that were injected with food dye (right), but not when food dye is offered in a cup for the same amount of time (left). This shows that food dye itself is not attractive to the ants. Two-sided Mann–Whitney test, $U = 100$, ***$P = 2.42E-05$. n = 10 replicate colonies of 10 adults and 10 pupae each. Box plots with center line (median), box limits (interquartile range: IQR), and whiskers ($1.5 \times$ IQR).

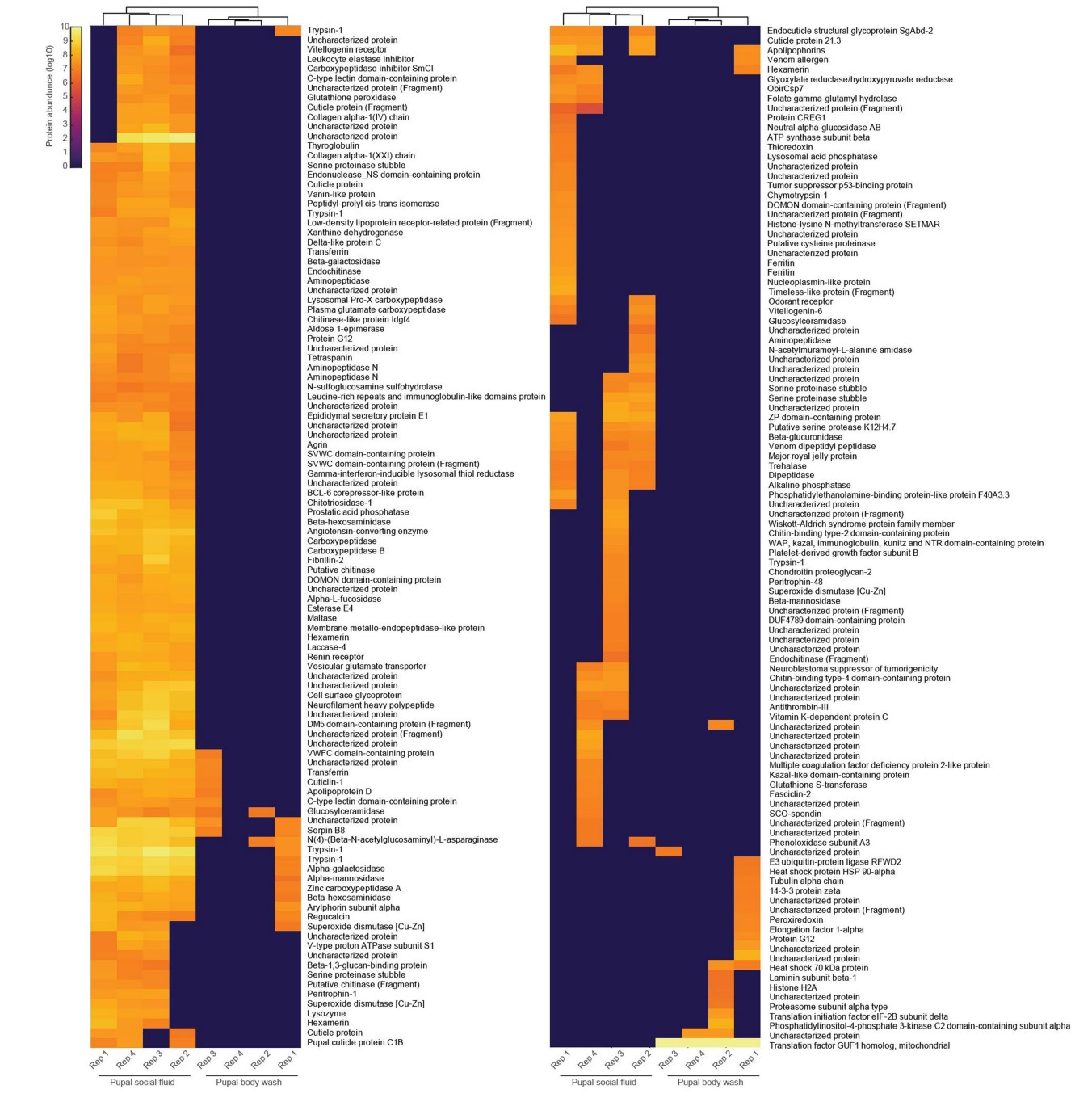

**Extended Data Fig. 2 | Proteomic profiles of pupal social fluid and whole-body wash.** Data are from n = 4 biological replicates of 426–493 pupae each. Samples collected on days 1–5 before eclosion were pooled. Phylogenies above columns represent similarities between samples.

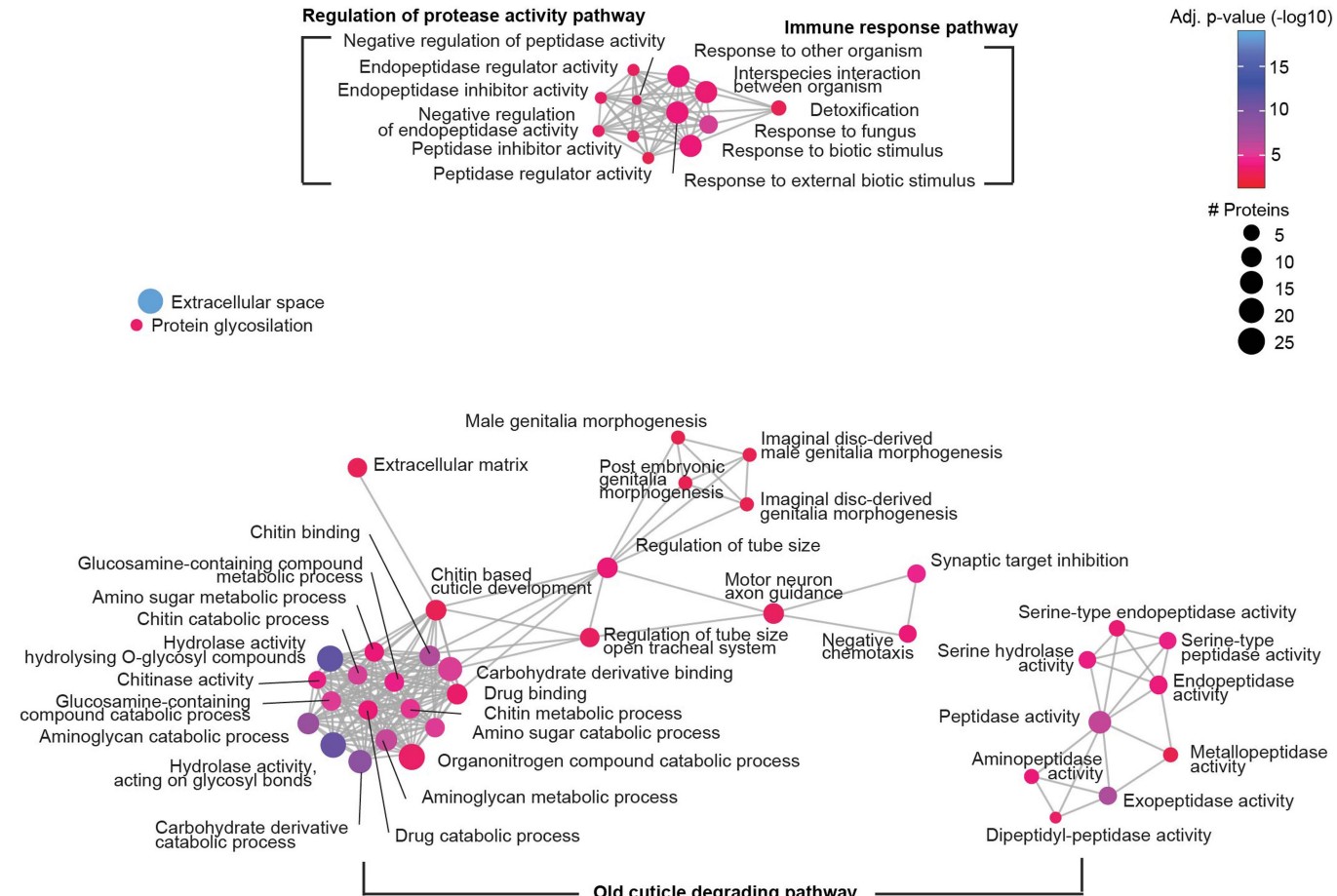

**Extended Data Fig. 3 | Proteomic GO enrichment analysis of pupal social fluid.** Nodes correspond to enriched GO terms (one-sided hypergeometric test, FDR = 0.05), node sizes indicate protein numbers in enriched GO terms, and edges are connecting overlapping protein sets.

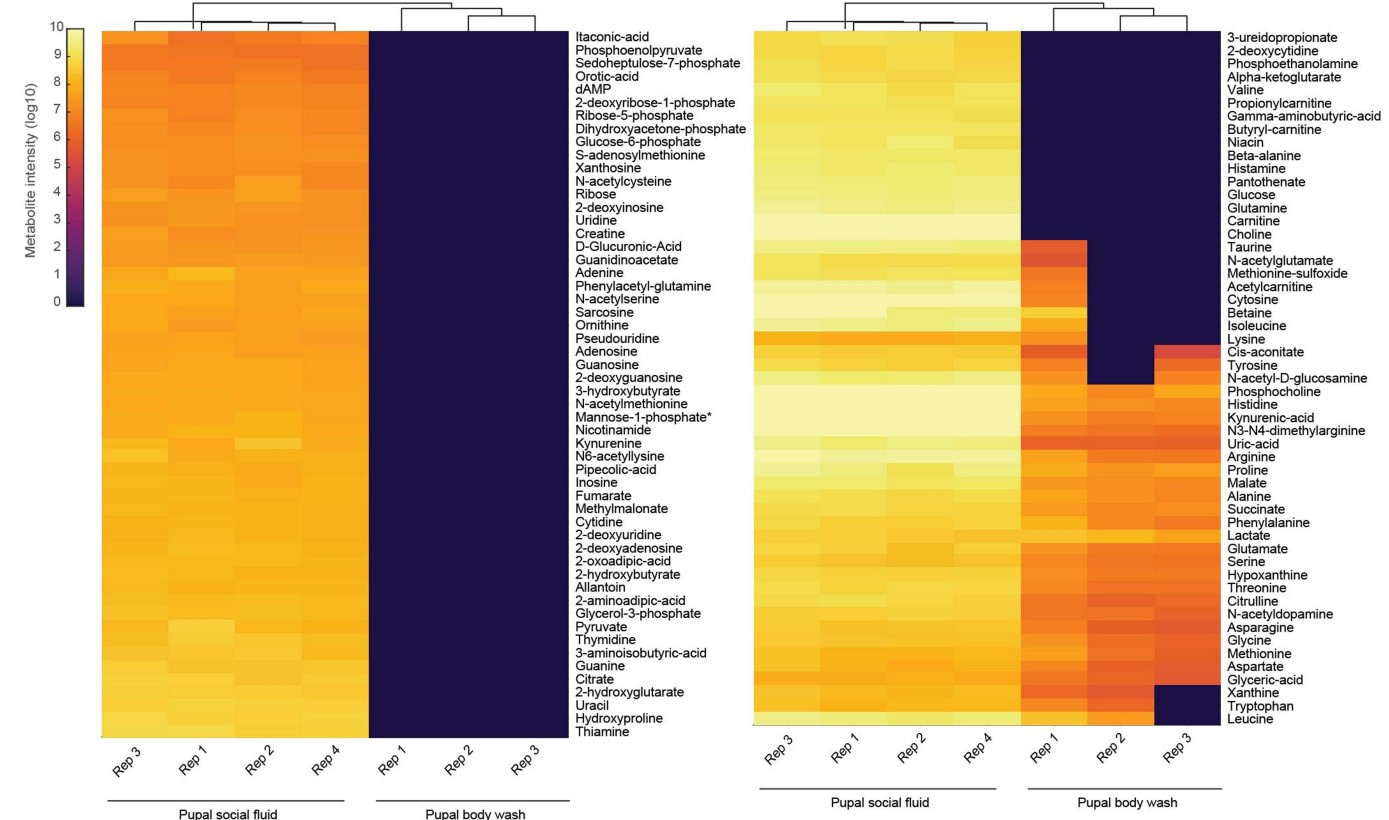

**Extended Data Fig. 4 | Metabolomic profiles of pupal social fluid and whole-body wash.** Data are from n = 4 biological replicates of 426–493 pupae each. Samples collected on days 1–5 before eclosion were pooled. Phylogenies above columns represent similarities between samples.

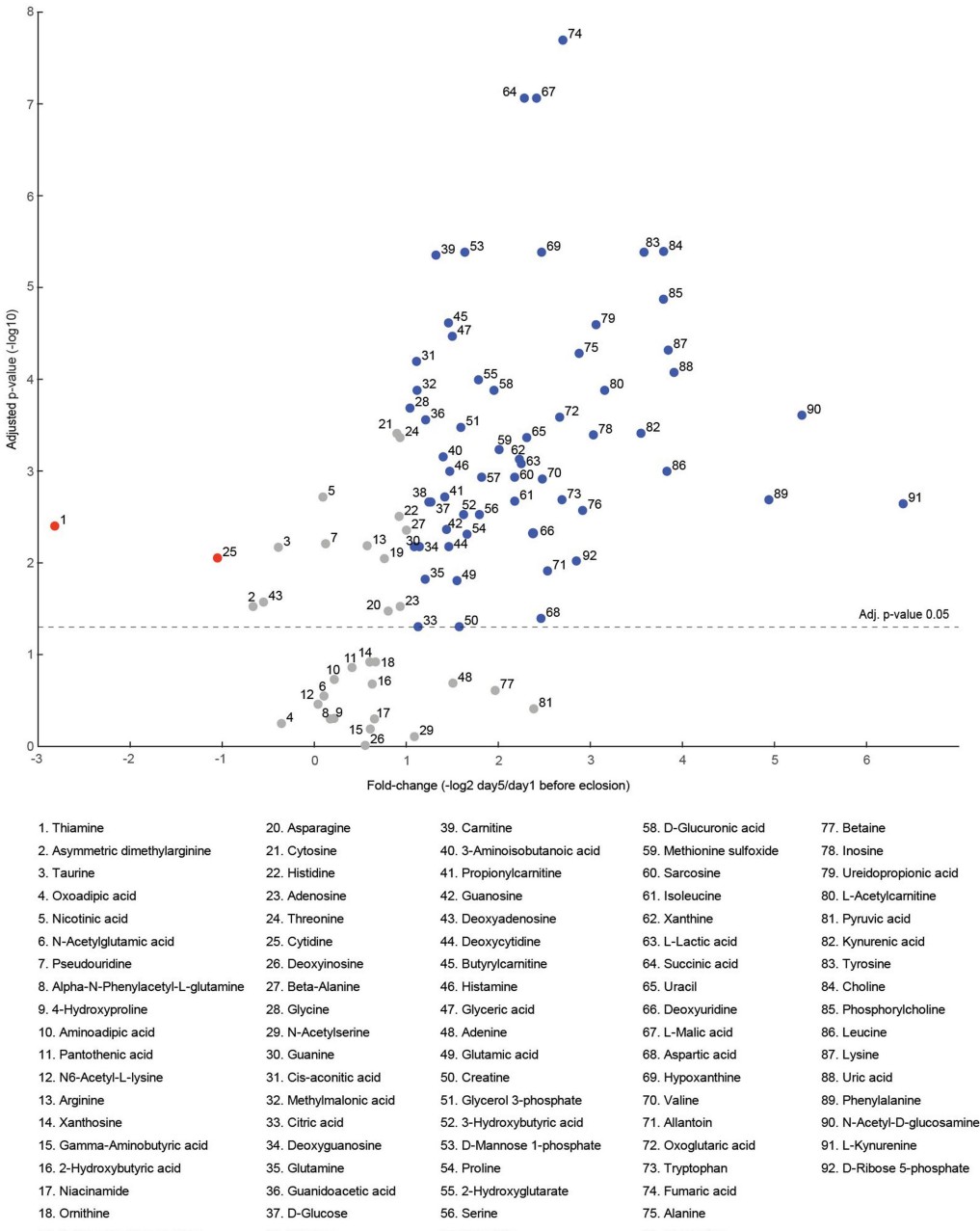

1. Thiamine
2. Asymmetric dimethylarginine
3. Taurine
4. Oxoadipic acid
5. Nicotinic acid
6. N-Acetylglutamic acid
7. Pseudouridine
8. Alpha-N-Phenylacetyl-L-glutamine
9. 4-Hydroxyproline
10. Aminoadipic acid
11. Pantothenic acid
12. N6-Acetyl-L-lysine
13. Arginine
14. Xanthosine
15. Gamma-Aminobutyric acid
16. 2-Hydroxybutyric acid
17. Niacinamide
18. Ornithine
19. O-Phosphoethanolamine

20. Asparagine
21. Cytosine
22. Histidine
23. Adenosine
24. Threonine
25. Cytidine
26. Deoxyinosine
27. Beta-Alanine
28. Glycine
29. N-Acetylserine
30. Guanine
31. Cis-aconitic acid
32. Methylmalonic acid
33. Citric acid
34. Deoxyguanosine
35. Glutamine
36. Guanidoacetic acid
37. D-Glucose
38. Citrulline

39. Carnitine
40. 3-Aminoisobutanoic acid
41. Propionylcarnitine
42. Guanosine
43. Deoxyadenosine
44. Deoxycytidine
45. Butyrylcarnitine
46. Histamine
47. Glyceric acid
48. Adenine
49. Glutamic acid
50. Creatine
51. Glycerol 3-phosphate
52. 3-Hydroxybutyric acid
53. D-Mannose 1-phosphate
54. Proline
55. 2-Hydroxyglutarate
56. Serine
57. Thymidine

58. D-Glucuronic acid
59. Methionine sulfoxide
60. Sarcosine
61. Isoleucine
62. Xanthine
63. L-Lactic acid
64. Succinic acid
65. Uracil
66. Deoxyuridine
67. L-Malic acid
68. Aspartic acid
69. Hypoxanthine
70. Valine
71. Allantoin
72. Oxoglutaric acid
73. Tryptophan
74. Fumaric acid
75. Alanine
76. Methionine

77. Betaine
78. Inosine
79. Ureidopropionic acid
80. L-Acetylcarnitine
81. Pyruvic acid
82. Kynurenic acid
83. Tyrosine
84. Choline
85. Phosphorylcholine
86. Leucine
87. Lysine
88. Uric acid
89. Phenylalanine
90. N-Acetyl-D-glucosamine
91. L-Kynurenine
92. D-Ribose 5-phosphate

**Extended Data Fig. 5 | Metabolite intensity of time series samples (1–5 days before eclosion).** Normalized to volume secreted/day/pupa from n = 4 biological replicates with 426–493 pupae each. Adjusted p-values obtained using one-way repeated measures ANOVA with auto scaling normalization, FDR 0.05. Fold-change values are between replicate averages from days 5 and 1 before eclosion. Metabolites with |fold-change| > 2 and significant changes in intensity across time are in red (decrease) and blue (increase). Metabolites with no significant change or |fold-change| < 2 are in gray.

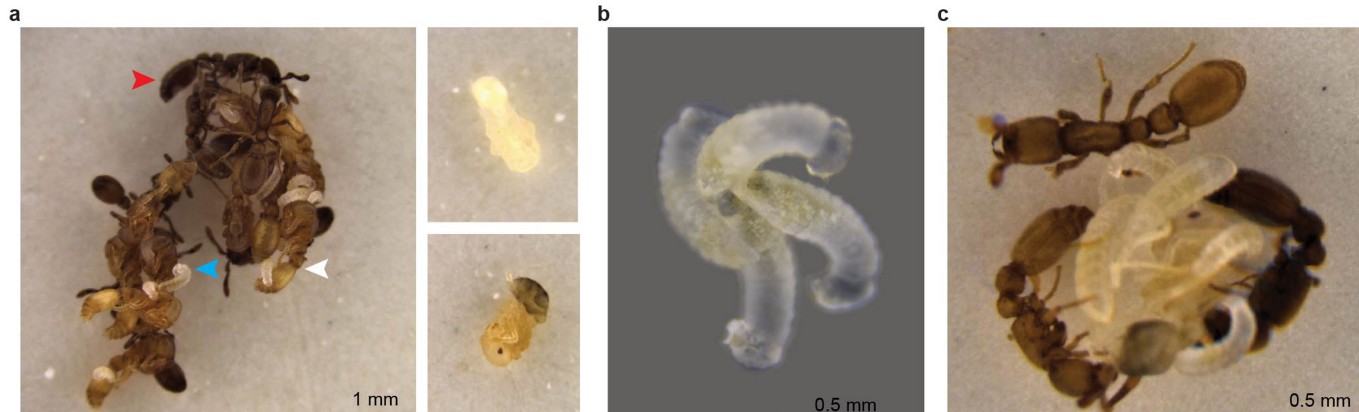

**Extended Data Fig. 6 | Adult preference for placing young larvae on alternative food sources.** (**a**) Colonies of 10 adults and 10 young larvae were given 10 *O. biroi* pupae and 10 prey items (dead *S. invicta* pupae). Adults placed early instar larvae on pupae (left), while prey items were left untouched and intact (right; 2 representative examples from the same colony). Arrowheads in left image indicate a pupa (white), a larva (blue) and an adult (red). (**b**) When colonies of 10 Adults and 10 young larvae received 10 prey items only, larvae were often arranged in tight clusters, but not placed on prey. (**c**) Adults readily place later instar larvae on prey items (here a dead *S. invicta* pupa). The figure shows representative images from experiments with n = 8 replicate colonies.

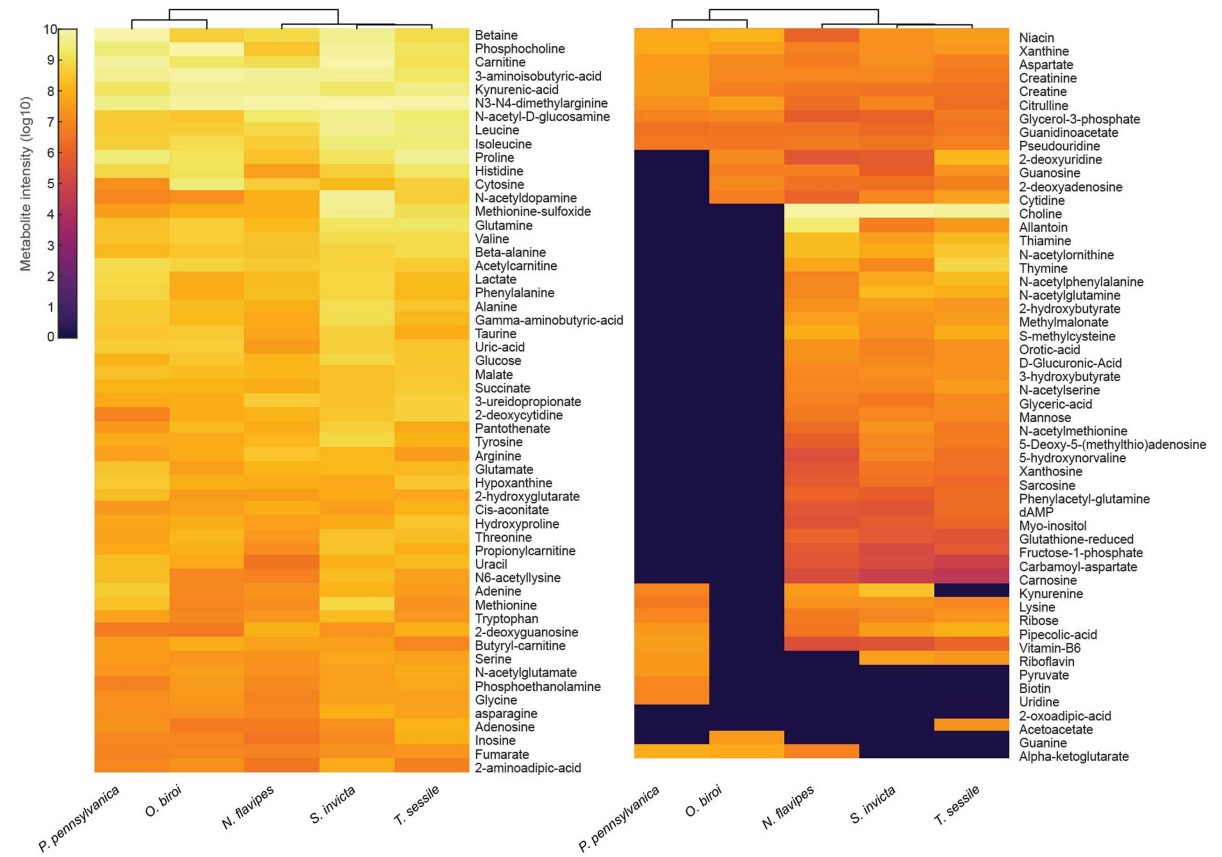

**Extended Data Fig. 7 | Metabolomic profiles of pupal social fluid of the species *Ponera pennsylvanica, Ooceraea biroi, Nylanderia flavipes, Solenopsis invicta* and *Tapinoma sessile*.** Data are from n = 1 biological replicates with 30 pupae each. Samples collected on days 1–4 before eclosion were pooled. Phylogenies next to columns represent similarities between species. (*)HMDB0006330 mannose-1-phosphate and HMDB0001078 mannose-6-phosphate could not be differentiated.

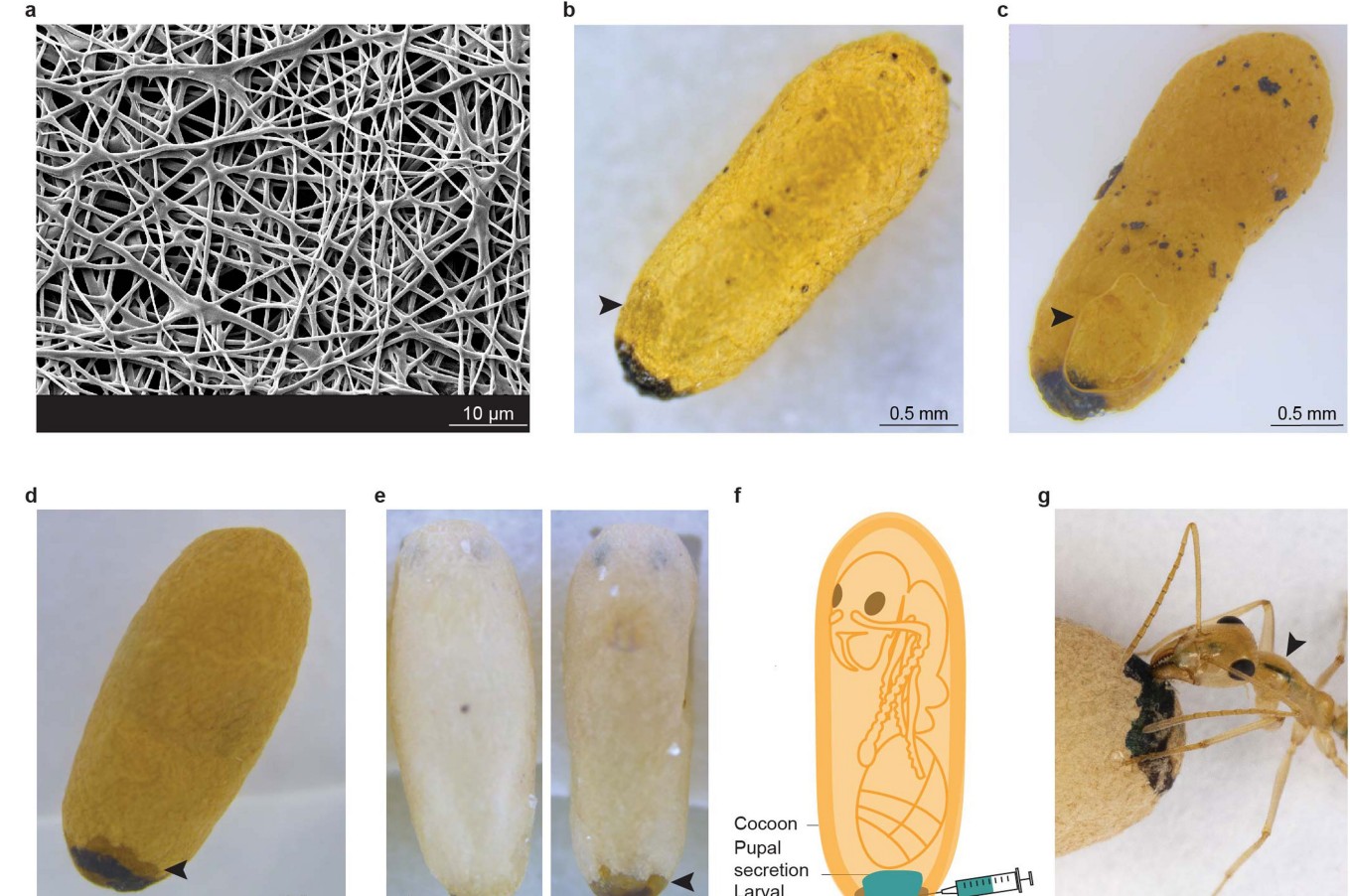

**Extended Data Fig. 8 | In species with enclosed pupae, workers consume pupal secretions directly from the cocoon.** (**a**) A scanning electron microscopy image of a *P. pennsylvanica* cocoon illustrates its porous fabric. n = 1 cocoon was imaged. Similar images of *Odontomachus brunneus* (subfamily Ponerinae) cocoons have been published elsewhere[40]. (**b**) A *P. pennsylvanica* cocoon after 1 day in social isolation. The arrowhead indicates the accumulation of pupal fluid. (**c**) Gently touching a glass slide to the surface of a *P. pennsylvanica* cocoon after 1 day in social isolation produces a droplet of pupal fluid on the underside of the slide (arrowhead). This illustrates that the pupal fluid readily crosses the silken fabric of the cocoon. (**d**) Same as in (b) for a cocoon of *Lasius neoniger* (subfamily Formicinae). (**e**) A young white pupa (left) and an older melanized pupa (right) of *Myrmecocystus mexicanus* (subfamily Formicinae) enclosed in cocoons after 1 day in social isolation. The arrowhead indicates the accumulation of pupal fluid in the melanized pupa, but not the white pupa. (**f**) Schematic of dye injection into pupal fluid in species where pupae are enclosed in a cocoon. (**g**) A *M. mexicanus* worker drinking dyed pupal fluid from a cocoon. The arrowhead indicates ingested blue dye visible through the translucent cuticle. The dark black spots at the bottom of the cocoons in (b–g) are the larval meconia, i.e., the metabolic waste products expelled by ant larvae as they enter pupation. The experiments and observations in (b–e, g) were repeated with at least n = 3 cocoons, yielding consistent results. Representative images are shown here.

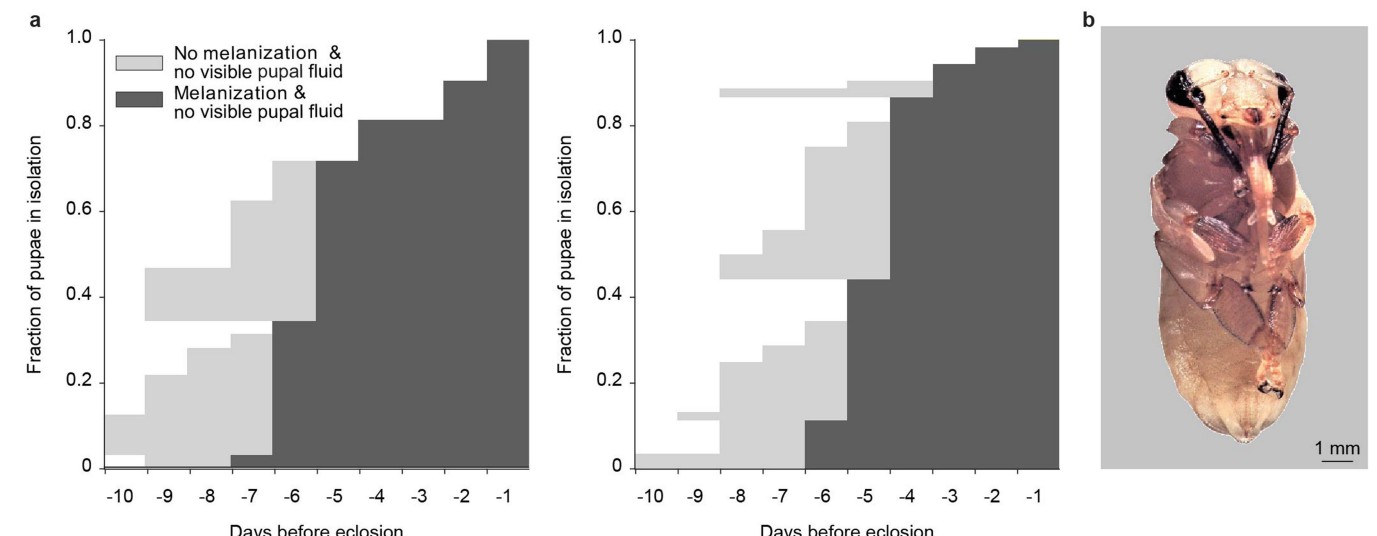

**Extended Data Fig. 9 | Honeybees do not secrete fluids during pupal development. (a)** Patterns of melanization without visible secretion of individually isolated pupae during development. Pupae of unknown ages were socially isolated and reared until eclosion. Pupae were either reared in 100% relative humidity to replicate conditions used for the different ant species (n = 32 pupae; left), or at 75% relative humidity as recommended in the literature[30] (n = 52 pupae; right). Both conditions had high survival rates, with 87% and 81% of pupae surviving to the adult stage at 100% and 75% relative humidity, respectively. Sample sizes and figure panels only include pupae that survived to eclosion. **(b)** *A. mellifera* pupa reared in social isolation. Shown is a representative image from the experiment in (a).

# Reporting Summary

## Statistics

For all statistical analyses, confirm that the following items are present in the figure legend, table legend, main text, or Methods section.

| n/a | Confirmed | |
|---|---|---|
| ☐ | ☒ | The exact sample size (*n*) for each experimental group/condition, given as a discrete number and unit of measurement |
| ☐ | ☒ | A statement on whether measurements were taken from distinct samples or whether the same sample was measured repeatedly |
| ☐ | ☒ | The statistical test(s) used AND whether they are one- or two-sided<br>*Only common tests should be described solely by name; describe more complex techniques in the Methods section.* |
| ☒ | ☐ | A description of all covariates tested |
| ☐ | ☒ | A description of any assumptions or corrections, such as tests of normality and adjustment for multiple comparisons |
| ☐ | ☒ | A full description of the statistical parameters including central tendency (e.g. means) or other basic estimates (e.g. regression coefficient) AND variation (e.g. standard deviation) or associated estimates of uncertainty (e.g. confidence intervals) |
| ☐ | ☒ | For null hypothesis testing, the test statistic (e.g. *F*, *t*, *r*) with confidence intervals, effect sizes, degrees of freedom and *P* value noted<br>*Give P values as exact values whenever suitable.* |
| ☒ | ☐ | For Bayesian analysis, information on the choice of priors and Markov chain Monte Carlo settings |
| ☒ | ☐ | For hierarchical and complex designs, identification of the appropriate level for tests and full reporting of outcomes |
| ☐ | ☒ | Estimates of effect sizes (e.g. Cohen's *d*, Pearson's *r*), indicating how they were calculated |

*Our web collection on statistics for biologists contains articles on many of the points above.*

## Software and code

Policy information about availability of computer code

| Data collection | VisiView acquisition software v 4.5.0.13 (VisiTech International, UK)<br>Leica Application Suite v 4.12.0 (Leica Microsystems, Switzerland) |
|---|---|
| Data analysis | MATLAB R2020b (MathWorks)<br>R v 3.6.3<br>Proteome Discoverer v.1.4 (Thermo Scientific)<br>EggNog-Mapper tool (http://eggnog-mapper.embl.de, emapper version 1.0.3-35-g63c274b, EggNogDB version 2)<br>clusterProfiler package (R)<br>Skyline Daily v 20.1 (MacCoss Lab)<br>MetaboAnalyst v 4.0<br>ImageJ/Fiji v 1.52p |

For manuscripts utilizing custom algorithms or software that are central to the research but not yet described in published literature, software must be made available to editors and reviewers. We strongly encourage code deposition in a community repository (e.g. GitHub). See the Nature Portfolio guidelines for submitting code & software for further information.

## Data

Policy information about availability of data

All manuscripts must include a data availability statement. This statement should provide the following information, where applicable:

- Accession codes, unique identifiers, or web links for publicly available datasets
- A description of any restrictions on data availability
- For clinical datasets or third party data, please ensure that the statement adheres to our policy

> The main data supporting the findings of this study are available within the paper and its Supplementary Information. Mass spectrometry proteomics data have been uploaded to the PRIDE database (accession number PXD037344).
> Databases used in the study are UniProtKB https://www.uniprot.org/ and HMDB https://hmdb.ca/

# Field-specific reporting

Please select the one below that is the best fit for your research. If you are not sure, read the appropriate sections before making your selection.

☒ Life sciences          ☐ Behavioural & social sciences          ☐ Ecological, evolutionary & environmental sciences

For a reference copy of the document with all sections, see nature.com/documents/nr-reporting-summary-flat.pdf

# Life sciences study design

All studies must disclose on these points even when the disclosure is negative.

| | |
|---|---|
| Sample size | Sample size calculations were not performed for the experiments. Appropriate sample sizes were estimated based on preliminary data that showed strong effects with small variations. |
| Data exclusions | No data was exluded. |
| Replication | Replication was done using individuals from different stock colonies at different dates to minimize batch effects. All attempts of replication were successful. All experiments were replicated three times or more, except the metabolomic profiling in Fig. 4b in which each sample contained secretion from 30 individuals that were pooled, as well as the scanning electron microscopy image of a P. pennsylvanica cocoon in Extended Data Fig. 8a. |
| Randomization | Experimental ants were collected haphazardly from stock colonies and distributed across experimental colonies/setups at random. The order of collection and place in which the colonies/setups were arranged were randomized. |
| Blinding | Blinding was not applicable in this study due to the descriptive nature of the experiments. |

# Reporting for specific materials, systems and methods

We require information from authors about some types of materials, experimental systems and methods used in many studies. Here, indicate whether each material, system or method listed is relevant to your study. If you are not sure if a list item applies to your research, read the appropriate section before selecting a response.

## Materials & experimental systems

| n/a | Involved in the study |
|---|---|
| ☒ | ☐ Antibodies |
| ☒ | ☐ Eukaryotic cell lines |
| ☒ | ☐ Palaeontology and archaeology |
| ☐ | ☒ Animals and other organisms |
| ☒ | ☐ Human research participants |
| ☒ | ☐ Clinical data |
| ☒ | ☐ Dual use research of concern |

## Methods

| n/a | Involved in the study |
|---|---|
| ☒ | ☐ ChIP-seq |
| ☒ | ☐ Flow cytometry |
| ☒ | ☐ MRI-based neuroimaging |

## Animals and other organisms

Policy information about studies involving animals; ARRIVE guidelines recommended for reporting animal research

| | |
|---|---|
| Laboratory animals | The study included one to several months old Ooceraea biroi females, as well as females of different developmental stages (larvae and pupae). Only female ants display social behavior, and all worker ants are female. In O. biroi in particular, males are only produced sporadically and do not partake in the social life of the colony. Conducting experiments with males is therefore not relevant in the context of the current study. |

| Wild animals | *Provide details on animals observed in or captured in the field; report species, sex and age where possible. Describe how animals were caught and transported and what happened to captive animals after the study (if killed, explain why and describe method; if released, say where and when) OR state that the study did not involve wild animals.* |
|---|---|
| Field-collected samples | Nylanderia flavipes, Tapinoma sessile, Ponera pennsylvanica, Solenopsis invicta, Lasius neoniger and Myrmecocystus mexicanus (no ethics guidelines apply). |
| Ethics oversight | No ethical approval was required since the animals used in this study are insects. |

Note that full information on the approval of the study protocol must also be provided in the manuscript.

