## [Peer Review File · Nature]

Editorial Note: the final version was seen by our reviewers

Manuscript Title: The pupal molting fluid has evolved social functions in ants

Reviewer Comments & Author Rebuttals

Reviewer Reports on the Initial Version:

Referees' comments:

Referee #1 (Remarks to the Author):

In insects with complete metamorphosis, pupae represent the resting stage between larvae and adults. While larvae actively move, feed or beg for food, pupae are typically inactive. This study uncovers an active role of ant pupae that appears to be important in social interactions within the colony.

Using the clonal raider ant as study organism, Snir and colleagues observed that pupae reared in isolation secrete fluid from the abdominal tip during the last 6 days before eclosion. They therefore designed a series of ingenious experiments to clarify the role of this pupal fluid. By injecting blue food dye underneath the pupal skin (exuvia), they found that 24 hours later adult ants had dye in their digestive system.

Behavioural observations confirmed that adult workers contact pupae frequently with their mouthparts and are attracted by the fluid. Pupae with experimentally occluded abdominal tip were less attractive to adult workers than control pupae. If the fluid is not removed from the pupae, they become contaminated with fungi and die. Adult workers place young larvae on the pupae (larvae cannot walk). The use of food dye, as above, confirmed that larvae consume the pupal fluid (dye staining found in larval gut). This fluid appears to be important especially for young larvae, while older larvae are placed by workers on prey items.

Another experiment showed that in the presence of pupae, which provide this milk-like fluid, larvae grow faster and survive better than in absence of pupae (but in presence of prey items), confirming the crucial role of the pupal fluid for colony development.

Proteomic and metabolomics profiling showed that the fluid is rich in nutrients, hormones and neuroactive substances.

The authors then tested whether this pupal fluid is specific of the clonal raider ant, *O. biroi* (subfamily Dorylinae), or widespread among ants, by isolating pupae of four additional species, in order to cover the five major ant subfamilies (Dorylinae, Myrmicinae, Formicinae, Dolichoderinae, Ponerinae).

Pupae of all these species secrete fluid droplets, which are similar in composition to those of *O. biroi* (as shown by metabolomics profiling). Experimental injection of dye in pupae of other species (*S. invicta*, Myrmicinae; *N. flavipes*, Formicinae) confirmed that adults and larvae consume the pupal fluid, as in *O. biroi*.

This original study thus describes and characterizes an overlooked but pivotal role of ant pupae in colony functioning. The results are of interest to many people working in Animal Behaviour and Social Evolution.

After careful scrutinizing, I could not find any methodological flaw in this study (I am not a specialist in metabolomics, therefore I cannot accurately evaluate the methodology used in this respect).

The behavioural experiments and manipulations are ingenious and carefully performed. Control treatments are appropriate.

Sample size is generally large. The statistical analysis is sound.

The results are not over-interpreted. The conclusions follow the results.

The manuscript is extremely clear and well written, engaging and easy to read, the iconography is outstanding and very clear.

Page 3, line 30: it is Fig 1b and not 1a (melanized pupa).

I have only one question: many ant species, such as *O. biroi*, have naked pupae, but in several other ant species pupae build a cocoon around them. Indeed, at page 8, line 29-20 read "cocoon were removed in the case of *P. pennsylvanica*", which is the only species tested in this study in which pupae have a cocoon. Now, in natural conditions, when the cocoon is not experimentally removed, how does the pupal fluid get out of the cocoon? The cocoon is closed and usually there is the larval meconium at the abdominal tip. This should be clarified.

Referee #2 (Remarks to the Author):

This is a highly original study that reveals that pupal secretions, which appear to have homologous origins to pupal molting fluid, have important, previously unknown social functions in ants. The history of modern research on ant social behavior spans more than 100 years, but this discovery represents a novel step forward. Although the authors correctly positioned this finding as one of many surprising new discoveries anticipated by Hölldobler and Wilson (1990), I would like to add that this is an excellent study that continues the viewpoint of Wheeler (1910), who believed that trophic circulation within a society is an important factor in the evolution of social behavior in ants.

The research is excellent, but admittedly I have some doubts about the general importance of the social function of pupal fluid as the authors assumed.

Three major concerns

First. In many ants belonging to Formicinae and Ponerinae, the pupa is surrounded by a cocoon. How are pupal secretions handled by these ants? The authors say that they artificially removed the

cocoons of *Ponera pennsylvanica* and observed them, but there is no video or other information available on *Ponera*. This seems odd. I would imagine that the cocoon interferes with contact between pupal secretions and nestmates, so an explanation is needed. I imagine that in many ants, pupae are devoured by care workers, which may have something to do with consumption of the pupal fluid.

Second. In ants, generally incipient founding colonies and in species that stop reproducing in winter and have no brood present in winter, there are time periods when there are no pupae present in the colony. However, the lack of availability of the pupal fluid in these colonies does not cause the colony to die. In fact, even in *Oocerea biroi* the first cohort of eggs of brood-removed adults group hatch without pupae, but from what I have seen, they grow normally. Thus it is clear that pupal fluid, even if it has an important social function in ants as the authors correctly positioned it (providing quicker growth and high survival for young larvae), is not an essential mechanism for survival for other colony members except pupae themselves (to which removal is essential). Such is not the case for mammalian milk, which is an essential part of the obligatory supply of milk to the offspring if the mother is present. Comparing ant pupal fluid to milk (pages 2 and 6) may be an exaggeration. The importance of the fluid to the colony, if not essential, can be easily demonstrated in the following group rearing experiments. I wonder if the authors have comparative data with constant colony size, for example, if there is a difference in new worker production rate per worker between the first cohort of colonies after brood removal (pupal fluid not available) and the second and subsequent cohorts (pupal fluid is available) in *O. biroi*. Such data, if available, would strengthen the authors' point of the colony-level benefits from the pupal fluid.

Last. What body part of the pupa the secretion is excreted? Is there a special tissue in the tip of abdomen? Since no fluid leaks out when the pupal skin is punctured by injection, there must be a special duct. This is biologically important, so if unknown, please specify that this is unknown. It is known that some "primitive" ants have a similar socially circulated "colony blood" function, with larval body fluids (hemolymph) serving as food for the adults (Masuko 1986 Behavioral Ecology and Sociobiology 19: 249-255, 2019 Myrmecological News 29: 21-34). Some ant larvae have even been found to have specialized tissues for secreting body fluids (Masuko 1989 Behavioral Ecology and Sociobiology 24: 127-132). Although I am neither an expert in proteomics nor metabolomics techniques, it would be very interesting to see if pupal fluid bears any biochemical similarity to larval body fluid. Whether the biomechanical and biochemical properties of the social pupal fluid found in this study are more similar to the molting fluid of insects in general or to the larval body fluid that is exchanged for nutrients in some ants? This must be relevant to discussion on evolutionary origins of the pupal fluid.

Minor Comments.

The authors have suggested homology of the pupal fluid of ants with pupal molting fluid of solitary insects by proteomic and metabolomic profiling analyses. However, an appropriate non-ant outgroup would be required to accept the authors' idea that social functions of pupal fluid shown in this study are commonly derived traits associated with social evolution in ants. The information on solitary insects that recycle molting fluid before emergence presented in references 24 and 25 is phylogenetically remote. The authors' own observations in solitary Hymenoptera, such as parasitoid

wasps, would strengthen their conclusions, if available.

Statistics

The statistics used were modern standard methods and I do not see any problem with them. However, one thing the wording. The authors said that ants were sampled randomly. Randomization is easy for colonies. However, true randomized sampling is not easy when picking up individuals from a colony. The authors' sampling was probably a pseudorandom sampling (because it is not easy to assign an ID to all individuals in advance). How about changing the wording to such as "haphazardly sampled"?

L.39 "Whole-body wash" How this was collected? Did you remove pupae from each stock colony that was with the adults and wash them promptly?

Author Rebuttals to Initial Comments:

Referee #1 (Remarks to the Author):

This original study thus describes and characterizes an overlooked but pivotal role of ant pupae in colony functioning. The results are of interest to many people working in Animal Behaviour and Social Evolution.

After careful scrutinizing, I could not find any methodological flaw in this study (I am not a specialist in metabolomics, therefore I cannot accurately evaluate the methodology used in this respect).

The behavioural experiments and manipulations are ingenious and carefully performed. Control treatments are appropriate.

Sample size is generally large. The statistical analysis is sound.

The results are not over-interpreted. The conclusions follow the results.

The manuscript is extremely clear and well written, engaging and easy to read, the iconography is outstanding and very clear.

***We thank the reviewer for their enthusiasm and careful review of our manuscript.

Page 3, line 30: it is Fig 1b and not 1a (melanized pupa).

***The first statement in this sentence (“Secretion begins six days before eclosion, shortly after the pupae have melanized...”) refers to Fig. 1a, which shows the timecourse of pupal secretion in the lower part of the panel in blue. The second statement (“...forming a stereotypical droplet on the abdominal tip”) then refers to Fig. 1b, which shows a melanized pupa with an accumulated secretion droplet.

I have only one question: many ant species, such as *O. biroi*, have naked pupae, but in several other ant species pupae build a cocoon around them. Indeed, at page 8, line 29-20 read “cocoon were removed in the case of *P. pennsylvanica*”, which is the only species tested in this study in which pupae have a cocoon. Now, in natural conditions, when the cocoon is not experimentally removed, how does the pupal fluid get out of the cocoon? The cocoon is closed and usually there is the larval meconium at the abdominal tip. This should be clarified.

***This is an important point that we failed to clarify in the initial submission, and the same question was raised by Referee #2 (their first major concern below). The short answer is that the cocoon is permeable to the social fluid. We have now included several additional experiments, a new supplementary figure (Extended Data Fig. 8), and a new video (Supplementary Video 8) to illustrate this.

First, we performed scanning electron microscopy of *P. pennsylvanica* cocoons, showing that the silken fabric of ant cocoons is porous (Extended Data Fig. 8a; similar images of ant cocoons have been published in E.G.P. Fox, A.A. Smith, J.C. Gibson & D.R. Solis, Larvae of trap-jaw ants, *Odontomachus* LATREILLE, 1804 (Hymenoptera: Formicidae): morphology and biological notes. *Myrmecol. News.* **25**, 17-28 (2017)). Second, we placed enclosed pupae from three species in social isolation: *P. pennsylvanica* (subfamily Ponerinae), *Lasius neoniger* (Formicidae), and *Myrmecocystus mexicanus* (Formicidae). In each case, the accumulation of fluid in form of a dark wet spot at the abdominal tip can be observed within 24 hours (Extended Data Fig. 8b-e). The fluid easily penetrates the cocoon when gently appressed with a glass slide (Extended Data Fig. 8c). In *P. pennsylvanica* and *L. neoniger*, it is impossible to tell the developmental stage while the pupa is still enclosed in the cocoon. In *M. mexicanus*, however, the cocoon is sufficiently translucent to show that secretion only occurs in older, melanized pupae, but not in young pupae, the same pattern observed in other species (Extended Data Fig. 8e). Finally, we added small amounts of food dye to the pupal secretion of *M. mexicanus* by injecting directly through the cocoon (Extended Data Fig. 8f). Because the cuticle of *M. mexicanus* workers is translucent, this allowed us to directly visualize how ants drink the secretion from the intact cocoon (Extended Data Fig. 8g, Supplementary Video 8). In summary, this demonstrates that pupae enclosed in cocoons also secrete social fluid, and that adults consume this fluid, just like in species with naked pupae. Pupal social fluids thus indeed seem to be ubiquitous across the ants.

We describe these additional findings on lines 217-220 in the main manuscript as follows: “In species where pupae are enclosed in cocoons, such as *P. pennsylvanica*, adults consume the social fluid through the permeable silken fabric (Extended Data Fig. 8). This can be visualized

using dyes in ants with translucent cuticles, such as *Myrmecocystus mexicanus* (subfamily Formicinae) (Extended Data Fig. 8, Supplementary Video 8).”

We have also added the following sentence to the legend of Fig. 4 to avoid potential confusion (lines 712-713):

“In *P. pennsylvanica*, the cocoon has been removed to better show the secretion (also see Extended Data Fig. 8).”

Referee #2 (Remarks to the Author):

This is a highly original study that reveals that pupal secretions, which appear to have homologous origins to pupal molting fluid, have important, previously unknown social functions in ants. The history of modern research on ant social behavior spans more than 100 years, but this discovery represents a novel step forward. Although the authors correctly positioned this finding as one of many surprising new discoveries anticipated by Hölldobler and Wilson (1990), I would like to add that this is an excellent study that continues the viewpoint of Wheeler (1910), who believed that trophic circulation within a society is an important factor in the evolution of social behavior in ants.

***We thank the reviewer for their kind assessment and thoughtful comments. They are absolutely correct about the seminal contributions of W. M. Wheeler, whom we cited in the previous version but did not mention by name. We decided to add the following sentence to the introduction to highlight his role in our understanding of social insect trophallaxis: “William M. Wheeler, who coined the term, even suggested that trophallaxis would be the key to understanding eusocial behavior in insects (Wheeler 1918, 1923).” (Lines 47-48).

The research is excellent, but admittedly I have some doubts about the general importance of the social function of pupal fluid as the authors assumed.

***The reviewer raises several important points, and we have conducted additional experiments to address these concerns. Our results underscore how general and important the social function of pupal fluids is in ants.

Three major concerns

First. In many ants belonging to Formicinae and Ponerinae, the pupa is surrounded by a cocoon. How are pupal secretions handled by these ants? The authors say that they artificially removed the cocoons of *Ponera pennsylvanica* and observed them, but there is no video or other information available on *Ponera*. This seems odd. I would imagine that the cocoon interferes with contact between pupal secretions and nestmates, so an explanation is needed. I imagine that in many ants, pupae are devoured by care workers, which may have something to do with consumption of the pupal fluid.

***This same question was raised by Referee #1, and we provide a detailed response above. The cocoon does not constitute a barrier to the pupal secretion, as we now demonstrate with additional experiments.

Second. In ants, generally incipient founding colonies and in species that stop reproducing in winter and have no brood present in winter, there are time periods when there are no pupae present in the colony. However, the lack of availability of the pupal fluid in these colonies does not cause the colony to die. In fact, even in *Oocerea biroi* the first cohort of eggs of brood-removed adults group hatch without pupae, but from what I have seen, they grow normally. Thus it is clear that pupal fluid, even if it has an important social function in ants as the authors correctly positioned it (providing quicker growth and high survival for young larvae), is not an essential mechanism for survival for other colony members except pupae themselves (to which removal is essential). Such is not the case for mammalian milk, which is an essential part of the obligatory supply of milk to the offspring if the mother is present. Comparing ant pupal fluid to milk (pages 2 and 6) may be an exaggeration. The importance of the fluid to the colony, if not essential, can be easily demonstrated in the following group rearing experiments. I wonder if the authors have comparative data with constant colony size, for example, if there is a difference in new worker production rate per worker between the first cohort of colonies after brood removal (pupal fluid not available) and the second and subsequent cohorts (pupal fluid is available) in *O. biroi*. Such data, if available, would strengthen the authors' point of the colony-level benefits from the pupal fluid.

***The reviewer is correct that in incipient colonies or in colonies that overwinter without pupae, some larvae survive without access to pupal fluid. The same is true for *O. biroi*; colonies can “restart” in the absence of pupae. However, as we show in Fig. 3h, survival of larvae drops from over 90% to ca. 60% under these conditions. Therefore, even though pupal fluid is not strictly speaking “essential” for colony survival, the fitness effect at the colony level is dramatic (a ca. 30% reduction in colony growth). A lineage in which larvae never have access to pupal fluids would therefore be at a major disadvantage compared to lineages that feed larvae on pupal fluids. In other words, in the absence of pupal fluid, ants can rear young larvae on alternative / supplementary diets, but this is much less efficient. We discuss the question of what young ant larvae feed on, i.e., what these alternative diets might be in the Supplementary Discussion (interestingly, but maybe not surprisingly in light of the current study, not much is known about this issue; lines 839-847):

“What young ant larvae feed on is often not clear, and in several species, observations suggest that they do not receive the same diet of solid food as older larvae of the same species^{8,9,58} (Supplementary Video 5). Instead, in species with adult-to-larva trophallaxis such as *S. invicta*, it has been suggested that young larvae receive liquid food from workers via mouth-to-mouth feeding⁵⁹. In other species, young larvae consume trophic or even viable eggs^{8,60}. While there is no evidence for adult-to-larva trophallaxis in *O. biroi*, it is possible that young larvae in this species consume eggs, which are present during the colony cycle when larvae hatch²⁷. How exactly pupal secretions, eggs, adult trophallactic fluids, and external food sources contribute to early larval nutrition across the ant phylogeny should be a focus of future research.”

We thus agree with the reviewer that pupal fluids are not “essential” for larvae in the sense that absence of pupal fluids does not lead to 100% larval mortality (we have changed the one instance of “essential” in the manuscript to “important”, line 156). However, this is also not a defining feature of mammalian milk. Data from years of human infants fed cow milk and even plant-

based alternatives show that, although their consumption increases death rates, it does not cause 100% mortality (W.J. Howarth, The influence of feeding on the mortality of infants. *The Lancet*. **166**, 210-213 (1905); F. Falkner, *Infant and Child Nutrition Worldwide: Issues and Perspectives* (CRC Press, Boca Raton, FL, 1991)). Similarly, in other mammals, dietary alternatives do not necessarily lead to mortality but rather impaired growth (D. Huber, I. Kulier, A. Poljak, B. Devčić-Kuha, Food intake and mass gain of hand-reared brown bear cubs. *Zoo Biology*. **6**, 525-533 (1993); A.F. Kertz, T.M. Hill, J.D. Quigley, A.J. Heinrichs, J.G. Linn, J.K. Drackley, A 100-Year Review: Calf nutrition and management. *Journal of Dairy Science*. **12**, 10151-10172 (2017)). We therefore believe that our conclusion from our experiments, that they "...show that the pupal fluid serves as a milk-like substance for newly hatched larvae, greatly increasing larval growth and survival during the first days after hatching", is justified.

The reviewer also asked whether we had available data from a fitness experiment across multiple colony cycles. Unfortunately, we currently do not have such data. While the experiment suggested by the reviewer is interesting, it suffers from the fact that workers would be systematically younger in the first colony cycle without pupae than in later colony cycles with pupae. Because reproductive output in *O. biroi* declines as a function of age, this would confound the experiment. Because of this, in the experiment we report in the manuscript (Fig. 3g, h), we start with precisely standardized colonies that only differ in whether or not pupae are present and young larvae have access to pupal fluids. Because under natural conditions, *O. biroi* larvae only have access to pupal fluids during the first few days after hatching, we measure fitness effects during this period. The striking increase in larval mortality we observe in this experiment will almost certainly translate into a reduction in worker offspring and colony growth. After some consideration and in light of the arguments outlined here, we decided that the experiment suggested by the reviewer would not provide sufficient additional insights to justify the effort (the outlined experiment would have to be conducted over several colony cycles, i.e. would take at least four months to complete). However, if the reviewer and editor believe that such an experiment would in fact be essential we would be willing to reconsider.

Last. What body part of the pupa the secretion is excreted? Is there a special tissue in the tip of abdomen? Since no fluid leaks out when the pupal skin is punctured by injection, there must be a special duct. This is biologically important, so if unknown, please specify that this is unknown. It is known that some "primitive" ants have a similar socially circulated "colony blood" function, with larval body fluids (hemolymph) serving as food for the adults (Masuko 1986 *Behavioral Ecology and Sociobiology* 19: 249-255, 2019 *Myrmecological News* 29: 21-34). Some ant larvae have even been found to have specialized tissues for secreting body fluids (Masuko 1989 *Behavioral Ecology and Sociobiology* 24: 127-132). Although I am neither an expert in proteomics nor metabolomics techniques, it would be very interesting to see if pupal fluid bears any biochemical similarity to larval body fluid. Whether the biomechanical and biochemical properties of the social pupal fluid found in this study are more similar to the molting fluid of insects in general or to the larval body fluid that is exchanged for nutrients in some ants? This must be relevant to discussion on evolutionary origins of the pupal fluid.

***Where exactly the secretion comes from is an interesting question, which we have addressed with additional imaging experiments that are reported in a new supplementary figure (Extended Data Fig. 1). The pupal case has two openings on the abdominal tip, where the secreted fluid

accumulates, the genital opening and the rectal invagination (Extended Data Fig. 1a,b). Both directly connect the broad exuvial space to the outside. We have visualized the approximate volume and distribution of this space by air injection (Extended Data Fig. 1e). At least in *O. biroi*, the fluid droplet forms specifically from the rectal invagination (Extended Data Fig. 1c,d). It is possible that the genital opening facilitates this process by allowing an influx of air to replace the secreted fluid. It thus appears that the molting fluid, which is initially secreted from specialized epidermal cells and digests the old cuticle to form the exuvial space, accumulates inside the exuvial space before forming a droplet outside of the rectal invagination of the pupal case. The reason that no fluid leaks out after injections is that we use extra fine needles and inject very small volumes to minimize pressure. We report these additional findings on lines 69-70 in the revised manuscript as follows:

“The secretion initially accumulates in the exuvial space and then exudes from the rectal invagination of the pupal case (Extended Data Fig. 1)...” Additional details are provided in the supplementary figure.

The phenomenon of larval hemolymph feeding that has been described by Keiichi Masuko in a few ant species is certainly interesting and constitutes another fascinating example of social fluid exchange. Unfortunately, as far as we know, the composition of these fluids has never been analyzed, and meaningful comparisons are therefore currently impossible. The species for which this has been observed (genera *Leptanilla*, *Proceratium*, and *Stigmatomma*) are cryptic and live in small colonies. Collecting fluid quantities that would be suitable for proteomic or metabolomic analyses will therefore be a formidable challenge. However, we agree with the reviewer that this is a fruitful avenue for future investigations, and we hope that our study will help inspire such experiments.

Minor Comments.

The authors have suggested homology of the pupal fluid of ants with pupal molting fluid of solitary insects by proteomic and metabolomic profiling analyses. However, an appropriate non-ant outgroup would be required to accept the authors' idea that social functions of pupal fluid shown in this study are commonly derived traits associated with social evolution in ants. The information on solitary insects that recycle molting fluid before emergence presented in references 24 and 25 is phylogenetically remote. The authors' own observations in solitary Hymenoptera, such as parasitoid wasps, would strengthen their conclusions, if available.

***We conducted a literature search but could not find any pertinent reports about solitary hymenopterans. We therefore currently do not know whether some solitary hymenopterans secrete molting fluid. Either way, the social functions of pupal secretions we report here for ants must be derived. Almost by definition, pupal secretions in solitary insects cannot have the same social functions (there are no interactions across developmental stages), or lead to offspring death if not removed. Therefore, the conclusion referenced by the reviewer is not contingent on whether or not solitary Hymenoptera recycle or secrete molting fluid. However, we were intrigued by this comment and wondered whether other eusocial Hymenoptera might also have coopted the molting fluid as a social fluid. We therefore conducted an experiment with honeybees (which have evolved eusociality independently from ants), which then also gets at the question of whether other Hymenoptera recycle or secrete molting fluid. This experiment is

reported in the new Extended Data Fig. 9. It turns out that honeybee pupae do not secrete fluid, and that they eclose in social isolation outside of the comb without problems. We discuss this experiment in the context of the reviewer's comment on lines 222-230 as follows:

“While the social functions of the pupal secretion must be a derived trait in ants, it remains to be determined whether pupae of solitary hymenopterans produce secretions derived from the molting fluid. If so, these secretions might have readily been coopted during the various independent evolutionary origins of hymenopteran eusociality. We examined this possibility by socially isolating honeybee (*Apis mellifera*) pupae. None of the pupae produced visible secretion droplets (Extended Data Fig. 9), and over 80% of the pupae survived to eclosion in social isolation without additional assistance. While this shows that not all eusocial clades rely on the social functions of pupal fluids, the evolutionary dynamics of pupal secretions across the Hymenoptera will be a fruitful avenue for future investigation.”

Statistics

The statistics used were modern standard methods and I do not see any problem with them. However, one thing the wording. The authors said that ants were sampled randomly. Randomization is easy for colonies. However, true randomized sampling is not easy when picking up individuals from a colony. The authors' sampling was probably a pseudorandom sampling (because it is not easy to assign an ID to all individuals in advance). How about changing the wording to such as “haphazardly sampled”?

***We agree that "haphazardly sampled" more accurately describes the method used to collect individual ants from colonies. We have changed the wording in the reporting summary, which is where we mentioned random sampling, accordingly.

L.39 “Whole-body wash” How this was collected? Did you remove pupae from each stock colony that was with the adults and wash them promptly?

***That is correct. We have added information regarding the collection of whole-body wash samples on lines 270-273:

“While pupae were secreting, pupal whole-body wash samples were collected daily. The pupae were removed from colonies with adults and washed promptly with 1500 μ l LC/MS grade water. Whole-body wash samples were lyophilized and reconstituted in 15 μ l LC/MS grade water.”

Reviewer Reports on the First Revision:

Referees' comments:

Referee #1 (Remarks to the Author):

The Authors have carefully addressed the reviewers' comments and remarks by clarifying several important points and by conducting additional, pertinent, experiments.

They summarize these additional findings at the beginning of the rebuttal letter:

"We have conducted a series of additional experiments (...) These include various imaging and behavioral experiments to show how ants in species where pupae spin cocoons consume the pupal fluid. (...) and show that in honeybees, which represent an independent origin of hymenopteran eusociality, pupae do not secrete fluids. (...)"

Both reviewers believe that this study is original and represents a significant advance in our knowledge about social behavior in ants and -I'd like to add- about social dynamics in animals and the role of brood in social species in general.

I do not see any reason to further delay the acceptance of this manuscript.

Congratulations to the Authors for this nice piece of work.

I only have a small question for the Authors (I apologize in advance if the answer is already given somewhere in the MS or in one of the numerous additional files, I could not easily find it):

Do ants like food dye per se? I mean, if you give to some ant workers of the species tested in the experiments you report here a droplet of the food dye of the kind that you used, do they eat it? Is pure food dye palatable to the ants?

Minor comment: Extended data figure 8, please explain to the general reader what is a larval meconium.

All the best,
Patrizia d'Ettorre

Referee #2 (Remarks to the Author):

Dear Authors

The authors have responded honestly to all of my previous comments. Not only that, they have submitted several new data on a scale beyond my expectations. This, I believe, reinforces the

generality and importance of the authors' remarkable findings. There is nothing to further correct. This is a monumental paper in animal behavior and social evolution.

Sincerely
Kazuki Tsuji

Authors' Responses to 'Reviewer Reports on the First Revision':

1. *I only have a small question for the Authors (I apologize in advance if the answer is already given somewhere in the MS or in one of the numerous additional files, I could not easily find it): Do ants like food dye per se? I mean, if you give to some ant workers of the species tested in the experiments you report here a droplet of the food dye of the kind that you used, do they eat it? Is pure food dye palatable to the ants?*

This is a relevant point that we had failed to clarify. We have now included a control experiment in which we presented the ants either with a droplet of pure food dye in a small plastic cup, or with food dye injected into the exuvial space of pupae. Both treatments had 10 replicate colonies of 10 adults and 10 pupae each. After 24 hours, we dissected the adults' digestive systems and assessed dye staining, which would indicate that they had consumed the dye. All ants that were housed with injected pupae had consumed dye, while only two out of 100 ants that were kept with dye in a cup had consumed dye. This shows that ants are not attracted to pure food dye itself. We present these results in the new Extended Data Figure 1f.

2. *Minor comment: Extended data figure 8, please explain to the general reader what is a larval meconium.*

We have added the following explanation to the legend of Extended Data Figure 8: "The dark black spots at the bottom of the cocoons in (b-g) are the larval meconia, i.e., the metabolic waste products expelled by ant larvae as they enter pupation."